# Dissecting crosstalk induced by cell-cell communication using single-cell transcriptomic data

Jiawen Hou [1,2], Wei Zhao [1,2] ✉ & Qing Nie [1,2,3] ✉

During cell-cell communication (CCC), pathways activated by different ligand-receptor pairs may have crosstalk with each other. While multiple methods have been developed to infer CCC networks and their downstream response using single-cell RNA-seq data (scRNA-seq), the potential crosstalk between pathways connecting CCC with its downstream targets has been ignored. Here we introduce a machine learning-based method SigXTalk to analyze the crosstalk using scRNA-seq data by quantifying signal fidelity and specificity, two critical quantities measuring the effect of crosstalk. Specifically, a hypergraph learning method is used to encode the higher-order relations among receptors, transcription factors and target genes within regulatory pathways. Benchmarking of SigXTalk using simulation and real-world data shows the effectiveness, robustness, and accuracy in identifying key shared molecules among crosstalk pathways and their roles in transferring shared CCC information. Analysis of disease data shows SigXTalk's capability in identifying crucial signals, targets, regulatory networks, and CCC patterns that distinguish different disease conditions. Applications to the data with multiple time points reveals SigXTalk's capability in tracking the evolution of crosstalk pathways over time. Together our studies provide a systematic analysis of CCC-induced regulatory networks from the perspective of crosstalk between pathways.

Regulatory pathways in cells hardly function alone, and crosstalk often exists between multiple pathways. Such crosstalk may share upstream cell-cell communication (CCC) signals, intermediate regulators (e.g., transcription factors, TFs), and downstream target genes (TGs)[1–4]. The crosstalk between pathways can serve as useful functionalities in signal transduction[5]. For instance, Hippo signaling and TGF-β signaling could both be activated by the shared receptors *TGFBR1* and *TGFBR2*[6], and the activation and the inhibition between these pathways are carried out by phosphorylation or complex regulation mechanisms[7]. Different mitogen-activated protein kinase (MAPK) pathways may use a common kinase protein (e.g., *Ste11* or *Ste7* in *S. cerevisiae*)[8]. The expression of *Pax2* in *Drosophila* eyes is simultaneously regulated by both NOTCH

and EGFR pathways, neither of which is able to activate the shared target gene alone[6]. Due to the presence of shared signaling components (SSCs) that mediate the regulations, one pathway could be activated by the CCC signal of another pathway in the absence of its own signal, or may activate other target genes through signal leakage. Consequently, the expression pattern of the target genes is strongly affected by the crosstalk (Fig. 1a), which often integrates or re-allocates the signals from CCC. Depending on the regulations by the shared molecules, there are different types of crosstalk modules (Xmods) (Fig. 1b).

Reconstruction of cell-cell communication is the first step of analyzing crosstalk pathways, and a variety of computational methods

[1]Department of Mathematics, University of California Irvine, Irvine, CA, USA. [2]The NSF-Simons Center for Multiscale Cell Fate Research, University of California Irvine, Irvine, CA, USA. [3]Department of Developmental and Cell Biology, University of California Irvine, Irvine, CA, USA. ✉e-mail: zhaow17@uci.edu; qnie@uci.edu

have been developed to use single-cell RNA sequencing (scRNA-seq) data for CCC inference[9–12]. For instance, CellChat[13], Connectome[14] and CellPhoneDB[15] utilize the pre-curated ligand-receptor (LR) interaction databases and gene expression data to estimate the probability or strength of CCC from sender cells to receiver cells via the given LR pair. Information from auxiliary molecules and subunit structures is incorporated to increase the accuracy and reliability of CCC inference. Using spatial transcriptomics, CCC inference can be improved by

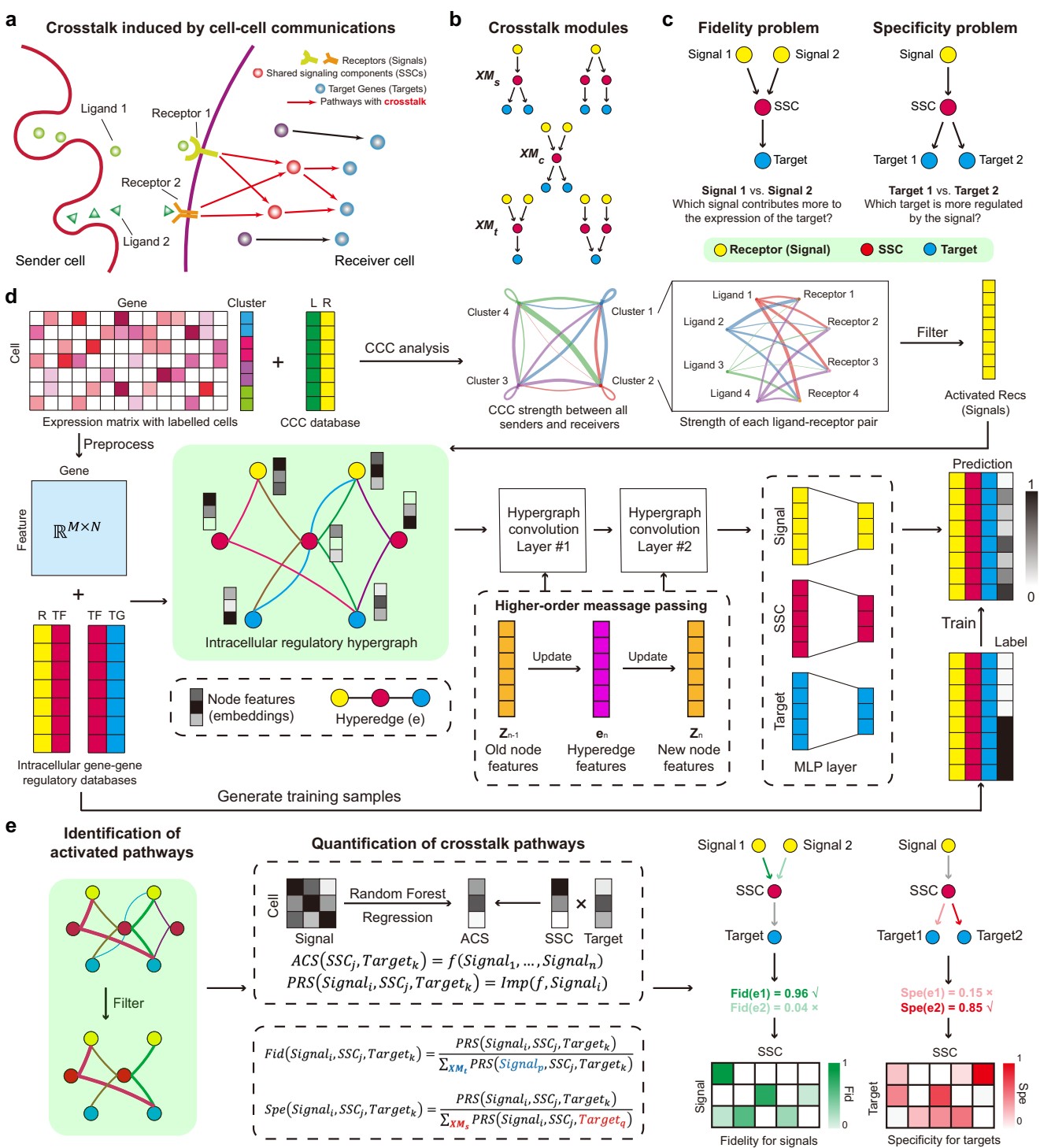

**Fig. 1 | Overview of SigXTalk. a** Receptors receive cell-cell communication (CCC) signals from sender cells, thus triggering a series of intracellular regulatory pathways. These pathways have crosstalk if they have the signals, SSCs or targets in common. **b** Crosstalk modules are identified and categorized depending on whether the pathways share signals (XM_s), SSCs (XM_C) or targets (XM_T). **c** The fidelity problem and specificity problem are raised to investigate the regulatory selectivity on signals and targets. **d** The workflow of identifying activated regulatory pathways. CCC analysis is performed to identify the activated receptors as signals. Then a hypergraph representation learning framework is employed to generate the low-dim features of genes and measure the activation probability of a pathway. **e** The identification and the quantification of the crosstalk pathways. The pathway regulatory strength is estimated by fitting the activation score of an SSC-Target pair using activated pathways, and the fidelity and specificity values are calculated using the proportion of a pathway's PRS within the crosstalk module it belongs to. L ligand, R receptor, TF transcription factor, SSC shared signaling component, TG target gene, Fid fidelity, Spe specificity.

considering the distance between sender and receiver cells. For example, Giotto[16] and SpaTalk[17] assume that the CCC only exists between sender cells and their neighbors, while SpaOTsc[18] and COMMOT[19] employ optimal transport-based approaches to infer CCC between individual cells within a certain distance.

To incorporate the downstream genes of CCC in the inference, several methods have been developed to construct a more comprehensive regulatory landscape from ligands in sender cells to target genes in receiver cells through analyzing gene regulatory networks (GRNs). CellCall[20] and scSeqComm[21] include the activity of transcription factors and connect CCC to the intracellular pathways, however, without characterizing the regulatory paths to individual target genes. GRN inference methods, like GRNBoost2[22], SCENIC[23], LEAP[24], GRISLI[25], Scribe[26] and GENELink[27], predict all the possible regulatory relationships within a set of genes (especially TFs and their corresponding TGs) using statistical, pseudo-time or deep learning techniques. Methods like NicheNet[28] and CytoTalk[29] use statistical approaches to identify activated LR pairs, and measure their effects on target genes in the receiver cells. StMLnet[30], Misty[31] and HoloNet[32] further integrate scRNA-seq data with spatial transcriptomics to improve the prediction of LR-TG relations by restricting the interaction of signaling molecules within a certain spatial distance.

However, existing methods are limited to measuring the direct and pairwise interactions between CCC signals and target genes based on users' specific choices, lacking a systematic analysis of intracellular signal transductions between multiple CCC signals and multiple targets. Most importantly, the current methods do not incorporate potential crosstalk that may lead to regulations from multiple signaling pathways. While some methods, such as BPLN[33], XTALK[34] and MuXTalk[35], use the pre-curated databases of cellular signaling pathways and employ the network-based algorithms and statistical tests to identify the significant crosstalk between pathways, those methods do not include single-cell gene expression data, can't provide cell type-specific crosstalk, and requires pre-choices of named pathways.

Given scRNA-seq datasets, is it possible to identify shared regulators or targets of multiple CCC signaling pathways? How to quantify the regulatory strength of pathways in the receiver cells when there is crosstalk between them? When a target gene is regulated by multiple pathways, what are the relative contributions of these pathways to the expression of the target gene? When a CCC signal regulates multiple target genes through different pathways, how is the regulatory effect allocated to each target gene? These questions can be phrased as "the fidelity problem" and "the specificity problem" (Fig. 1c). Here we introduce the concepts of pathway fidelity and specificity, which measure the "selectivity" between CCC signals and targets: fidelity of a pathway reflects its capacity to prevent activation of its target by CCC signals not intended for them, whereas specificity indicates the pathway's ability to avoid activating non-targets with its own CCC signal. Previously, using simple mathematical models, various mechanisms have been explored to analyze signal specificity and fidelity[1,36]. For example, insulating mechanisms such as blocking the potential crosstalk between pathways and spatial compartmentalization have been shown to maintain a high level of fidelity and specificity of multiple pathways[37,38].

To utilize single-cell gene expression data, here we develop SigXTalk, a computational method to analyze crosstalk between multiple CCC pathways using the concepts of fidelity and specificity. Through prior knowledge of gene-gene interaction, SigXTalk employs a specialized hypergraph learning framework to identify the crosstalk pathways and further measure their fidelity and specificity using tree-based machine learning approaches. Benchmark and applications to various scRNA-seq datasets suggest the capability and reliability of SigXTalk in systematic analysis of the relationship between the CCC signals and target genes that are connected via crosstalk.

## Results

### Overview of SigXTalk

SigXTalk requires the single-cell gene expression data and the cell cluster annotations as the input. The users can input specific genes of their interests, otherwise, SigXTalk uses the differentially expressed genes of each cell cluster as the target genes. The output includes the regulatory strength of all the possible pathways that originates from CCC signals and regulates the preset target genes, and the fidelity and specificity quantification of these pathways.

Specifically, a high fidelity of a pathway implies that the expression of its target gene is mainly regulated through this pathway instead of others, while a high specificity demonstrates that its target gene is likely the exclusive target of its CCC signal with little signal leakage to other targets. We first identify the crosstalk modules that are composed of all the pathways that share the same signaling molecules (see Methods for details). Then fidelity or specificity is calculated as the relative regulatory strength of a pathway within the crosstalk module, depending on the type of molecules pathways share.

In particular, SigXTalk performs the inference as follows (Fig. 1d, e). First, for the given receiver cells, the receptors that participate in cell-cell communication (CCC) with sender cells are firstly identified using CellChat. The activated receptors are recognized as signal inputs, while their downstream transcription factors (TFs) and target genes are named as shared signaling components (SSCs) and targets, respectively. Next, a hypergraph learning framework is employed to predict the regulatory relationships among them. To achieve this, a prior hypergraph skeleton is constructed using the most possible regulatory pathways based on pre-curated gene-gene interaction databases[28,39]. The constructed hypergraph along with processed gene expression matrix (as node features) are fed into a hypergraph neural network which contains a series of hypergraph convolution layers and linear layers. Through this, the expression of genes is encoded into a low-dimensional feature space while the information of their neighbors on the hypergraph is aggregated to incorporate the higher-order regulatory interactions. Once the hypergraph is trained using a supervised or self-supervised strategy, the output of this hypergraph learning framework is the predicted probability that a pathway is activated (Fig. 1d). After filtering out pathways with low probability, the pathway regulatory strength (PRS) of activated pathways is measured by regressing the activation level of an SSC-target pair using all the expression levels of its signals, which is performed using Random Forest, a tree-based statistical algorithm[40]. The pathway fidelity and specificity are calculated as the proportion of a given pathway regulatory strength within a crosstalk module, respectively (Fig. 1e, see Methods for details).

### Performance evaluation of SigXTalk

To investigate the performance of SigXTalk in identifying crosstalk pathways, we first use the simulated datasets with known ground truth. Specifically, we test whether SigXTalk can recover the activated regulatory pathways from incomplete observations of biological interactions. Learning from the scRNA data and the known gene-gene interactions (training samples), a good recovery tool is expected to accurately predict the unobserved regulatory pathways (test samples). To do so, we employ the Single-cell ExpRegression of Genes In silicO (SERGIO)[41] to simulate the single-cell gene expression profile guided by a hierarchical gene-gene interaction network that connects signals, SSCs and targets (see Methods for details). In this way, the simulated scRNA-seq data has its underlying regulatory landscape and the pre-defined network could be regarded as ground truth for the benchmarking task (Fig. 2a). The curve of the training/validation loss shows that SigXTalk's hypergraph learning framework can learn the useful regulatory information from training samples, represented by a drastic decay of training and validation loss within a few steps (Supplementary

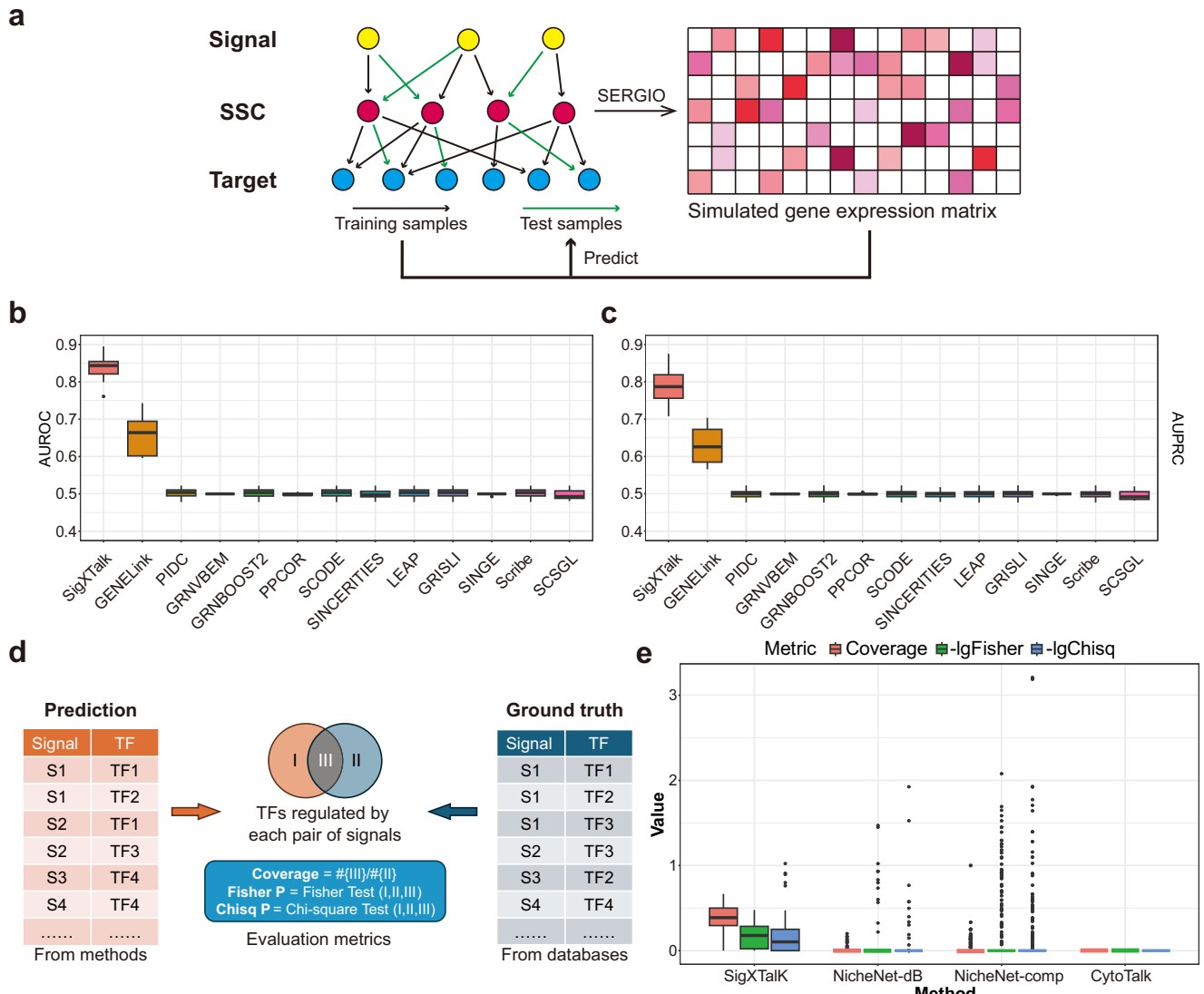

**Fig. 2 | Benchmarking SigXTalk against representative methods. a** The workflow of the benchmark using simulated data. **b** Boxplots for the AUROC of SigXTalk and 12 representative methods on 10 independent tests on the simulated data. **c** Boxplots for the AUPRC of SigXTalk and 12 representative methods on 10 independent tests on the simulated data. **d** The workflow of the benchmark using real data. **e** Boxplots for SigXTalk (42 identified receptor pairs), NicheNet-comp (366 identified receptor pairs), NicheNet-db (119 identified receptor pairs) and CytoTalk (0 identified receptor pairs), represented by the coverage, *p*-value of two-sided Fisher's exact test (shown in a log10 scale), and *p*-value of Chi-square test (shown in a log10 scale) for each receptor pair. Boxplot elements: center line, median; box limits, upper and lower quartiles; whiskers, 1.5x interquartile range; points, outliers. Source data are provided as a Source Data file.

Fig. 1a). We further compare its performance with 12 representative methods which are designed for recovering gene-gene links in GRNs, including GENELink, PIDC[42], GRNVBEM[43], GRNBOOST2, PPCOR[44], SCODE[45], SINCERITIES[46], LEAP, GRISLI, SINGE[47], Scribe[26] and SCSGL[48] (see Methods for details). By applying the 13 methods to simulated datasets, the probability that a potential combination of signal, SSC and target can form a regulatory pathway is obtained for each method on the test samples. AUROC and AUPRC for all these methods on 10 independent tests are presented (Fig. 2b, c). SigXTalk reaches the peak performance in recovering missing regulatory pathways, with a 18% increase in AUROC and 15% increase in AUPRC than the second-best method. These results imply that by considering the higher-order regulatory relationship, SigXTalk's hypergraph neural network can successfully identify the activated regulatory pathways, especially when there is crosstalk between them. However, the other methods consider only pairwise gene-gene interactions and underestimate the complex crosstalk between different pathways. For example, GENELink infers the GRN using an attention-based graph neural

network with a training strategy similar to SigXTalk, overlooking the SSCs' role in integrating multiple upstream regulatory information.

Next we study the robustness of SigXTalk by measuring its sensitivity against different conditions using different numbers of cells, numbers of genes, and crosstalk levels (Supplementary Note 1). SigX-Talk shows consistent performance in predicting activated pathways for datasets of different sizes (Supplementary Fig. 1b, c), and SigXTalk can be applied for identifying crosstalk across a wide range of crosstalk levels (Supplementary Fig. 1d). SigXTalk is relatively robust to the choice of hyperparameters, including the learning rate, the size of training batches, and the choice of random seed (Supplementary Fig. 1e−g).

We also study whether SigXTalk remains effective when the self-supervised training strategy is adopted, by using the highly-correlated gene pairs in the prior knowledge as the positive training set (Supplementary Note 1). Though the predicting accuracy gradually falls as the fraction of training samples drops, SigXTalk is still able to reach the AUROC value of 0.6 when only 20% of the regulatory pathways are

used for training (Supplementary Fig. 1h). In general, SigXTalk can leverage limited regulatory information, enabling the self-supervised training strategy to identify activated regulatory pathways using real-world scRNA-seq datasets.

The core concept of SigXTalk is the shared components in crosstalk pathways induced by cell-cell communication. To test whether SigXTalk can identify biologically meaningful shared components, we compare its performance with representative methods NicheNet, including using NicheNet prior ligand-target matrix, and CytoTalk, as they can estimate the target genes' response to CCC. Specifically, we study a dataset of head and neck squamous cell carcinoma (HNSCC)[49] that contains 9842 genes and 15,467 fully annotated cells. Based on the original study, the cells are grouped into eight major clusters with each cluster containing no fewer than 80 cells (Supplementary Fig. 2a). Using CAF cells as the receivers, for any pair of receptors in the receivers, each of the three methods generates a list of downstream TFs regulated by both receptors. We extract TFs that are shared by the same pair of receptors from the Kyoto Encyclopedia of Genes and Genomes (KEGG) databases[50]. The TFs with very low expression level or having low co-expression with receptors is filtered out. The predicted TF list is then compared to the ground truth TF list (Fig. 2d), and the performance is measured using three different metrics: the relative overlap of the two lists, the $p$-value of two-sided Fisher's exact test on the two lists, and the $p$-value of Chi-square test on the two lists (See Methods). SigXTalk can identify the shared TFs by the highest (36% on average) overlap with the potential shared TFs and higher confidence level than the other methods (Fig. 2e). Similar results are obtained when the methods are tested on the other four major cell types (Supplementary Table 1). Compared with methods, such as NicheNet that directly measures the regulatory relationship between ligands/receptors and targets, SigXTalk can more consistently predict the existence of intermediate regulators instead of simply recognizing them as targets. Compared with methods like CytoTalk that construct the whole inter- and intra-cellular signaling network regarding cell-cell communication, SigXTalk can involve more genes that are potential regulatory targets of receptors, along with their potential intermediate regulators.

Additionally, we test whether the selection of ground truth list may affect the performance of these methods. Specifically, we investigate the robustness of SigXTalk compared with the other two methods by varying the threshold of co-expression level between TFs and receptors. For this analysis, SigXTalk remains effective when only shared TFs co-expressed with upstream signals are selected as the ground truth. Filtering out the less co-expressed receptor-TF pairs can increase the coverage of SigXTalk by 20%, while other methods fail to improve the prediction performance (Supplementary Fig. 2b–e).

## Cell type specific crosstalk at the single-cell level

To illustrate how a target gene may be affected by its multiple upstream signals using the concept of fidelity, we consider the specific crosstalk between the canonical Wnt pathway (also known as the Wnt/β-Catenin pathway) and the non-canonical Wnt pathway (also known as the Wnt/PCP pathway). Generally, the two pathways have dual effects on each other's activity and the crosstalk dynamically regulates them to induce multiple processes like cell proliferation, migration, or differentiation[51–53]. Recently, it has been reported that a switch from canonical to non-canonical Wnt pathway triggers the differentiation from intestinal stem cells (ISCs) to Paneth progenitors using mouse's gut lineage dataset[54]. Naturally, the transcription factors are the SSCs, and interestingly, the differentiation process is associated with a change in the fidelity of the Wnt signals that regulate the biomarkers of ISCs and Paneth progenitors (Fig. 3a).

To quantify the crosstalk between canonical and non-canonical pathways, we first examine the Wnt pathways in the KEGG databases and extract the related Signal-SSC pairs using the Personalized

Pagerank (PPR) algorithm (see Methods)[55]. Using $Lgr5$ as the target, the PRS value of each possible Signal-SSC-Target pathway is directly estimated using the scRNA expression profile of ISC and Paneth progenitors. We find that, in ISCs, the expression of $Lgr5$ is mainly regulated by canonical signals $Lrp5$ and $Lrp6$, with the canonical-Wnt TF $Ctnnb1$. For Paneth progenitors, however, the regulation is dominated by non-canonical Wnt signals and their downstream TFs like Jun. A crosstalk between pathways is observed, represented by the regulation from the non-canonical co-receptor $Ryk$ to $Ctnnb1$ (Fig. 3b). The fidelity of each Wnt signal over all the possible SSCs is calculated, showing low fidelity of non-canonical in ISCs, and canonical Wnt signals in Paneth progenitors (Fig. 3c). These results are further validated, by directly estimating the activation of each pathway within each individual cell using the activation index (ACI, see Methods). Compared with non-canonical Wnt pathways, the ACI of canonical Wnt pathways are higher in ISCs, but lower in Paneth progenitors (Supplementary Fig. 3). In all, our findings agree with both results from individual-cell-based methods and previous experimental observations[54], demonstrating that the switch of Wnt pathways can be captured by SigXTalk.

## Quantifying the fidelity and specificity of crosstalk

As its basic functionality, SigXTalk identifies the activated regulatory pathways triggered by CCC events and quantifies the fidelity and specificity of crosstalk pathways. To demonstrate this function, we use CellChat to obtain various communication patterns targeting the malignant cells (Fig. 4a, b), such as $CD44$ as well as integrin signaling which have been reported to promote the metastasis of HNSCC[56]. $CD44$ and integrins are found to be the main receptors of the Collagen and Laminin CCC pathways, whose main senders include CAFs, Endothelial, and myofibroblasts (Supplementary Fig. 4a, b). Furthermore, we investigate if the malignant cells have heterogeneity in CCC patterns, by performing the subclustering and obtaining four subclusters (Supplementary Fig. 4c). CCC analysis using CellChat indicates that the 1st and 3rd subclusters of malignant cells show a slightly higher CCC activity in terms of the number as well as its total strength of CCC signaling (Supplementary Fig. 4d, e).

Most of the differentially expressed target genes of malignant cells have more than one crosstalk pathway. Specifically, while the majority of the number of crosstalk pathways of these genes falls in the interval from 1 to 50 (1322 out of 1873 selected targets), there are genes like epithelial tumor markers including cytokeratin, $EPCAM$ and $SFN$ having more than 100 activated pathways that regulate it, indicating a highly complex and integrated regulatory landscape (Fig. 4c, Supplementary Fig. 4f). To further reveal the various and complicated regulatory relationships, we look into the top 15 target genes with the most crosstalk pathways and investigate the contributions of CCC signals on the expressions of these genes (upper panel of Fig. 4d). Target genes like $KRT5$, $KRT14$, $KRT17$ are mainly regulated by $SDC1$, $DSG3$ and $CDH3$ but genes like $PRDX2$ and $RAN$ slightly prefer $BSG$, $CD44$, $CD47$ and $CD74$. On the contrary, although two genes in the integrin family (i.e. $ITGA3$ and $ITGB1$) participate in the CCC in malignant cells, their activities in regulating downstream targets show strong heterogeneity. For instance, $ITGB1$ regulates the expression of $CXCL14$ and $S100A10$ while $ITGA3$ only regulates $KRT14$. These results indicate different selectivity between CCC signals and target genes for crosstalk pathways. After a closer look at the crosstalk pathways that regulate the six tumor markers (Fig. 4e), we find that a signal can regulate multiple target genes, and a target gene can be regulated by multiple signals, via intertwined signal-TF-target pathways of different strength levels. For these target genes, $TNFRSF21$, $SDC1$, $DSG3$, $CDH3$, $CD74$ and $CD47$ account for the main signals. On the other hand, $FOS$, a proto-oncogene that drives the progression of epithelial cancer[57], is the main SSC where most of the regulations, regardless of the upstream signals, are realized through them (Fig. 4e, Supplementary

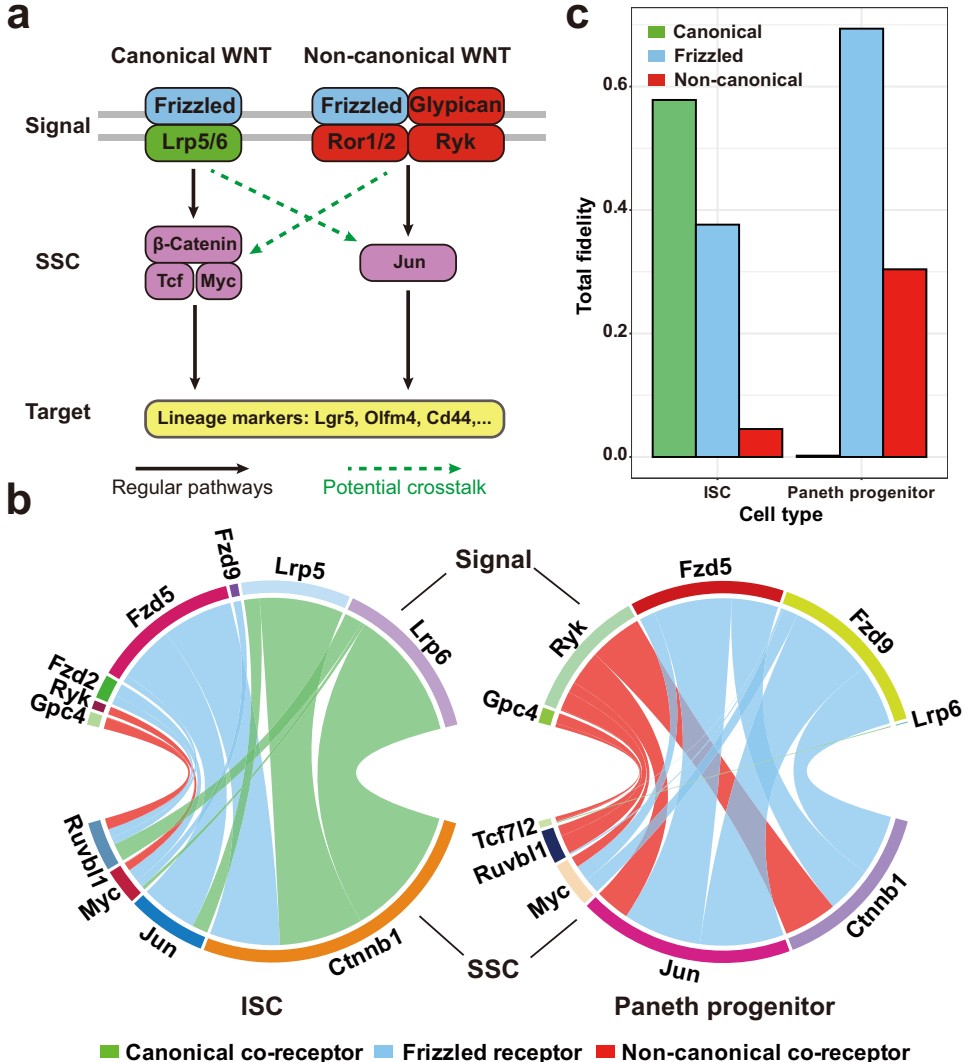

**Fig. 3 | Cell type-specific crosstalk in the context of a certain biological process. a** Overview of the canonical and non-canonical Wnt pathways during the lineage transition from intestinal stem cells (ISCs) to Paneth progenitors in mouse. The potential crosstalk between them may take part in the regulation of lineage markers; **b** The chord diagrams for the PRS values of the pathways regulating the target gene Lgr5. The width of the edge is proportional to the PRS value. Left: ISCs; Right: Paneth progenitors. **c** The bar charts for the total fidelity values of each kind of Wnt signal (canonical, non-canonical, or frizzled signals that are shared by both canonical and non-canonical pathways) in ISCs and Paneth progenitors. Source data are provided as a Source Data file.

Fig. 4g). Mechanically, *FOS* forms the transcription factor complex AP-1, which could be triggered by multiple CCC signals like *CD74* and *DSG3* by activating the MAPK pathway and/or suppressing the PPAR pathway[58,59]. The up-regulation of *FOS* subsequently induces the expression pattern of keratins[60]. Together SigXTalk identifies biologically meaningful pathways and regulators for this system.

Next, we investigate the fidelity and specificity measures of these pathways. Conceptually, fidelity and specificity delineate two facets of the regulatory pathway selectivity, i.e., selectivity on the signal versus selectivity on the target, respectively. Taking the target gene *KRT5* as an example, we calculated the fidelity distributions for each signal by evaluating fidelity values associated with different SSCs (Fig. 4f). Signals including *SDC1*, *DSG3* and *CDH3* show overall higher fidelity compared to other signals, indicating that the regulation of *KRT5* is mainly from these signals. A closer look at the combinations of signals and SSCs reveals that different SSC profiles are used to achieve high fidelity of different signals (Fig. 4g). For example, *DSG3* uses *FOS*, *PARK7* and *NFE2L2* to achieve high fidelity, *CDH3* uses *FOS*, *TP63* and *JUN*, and *SDC1* uses *FOS*, *PARK7*, *NFE2L2* and *EWSR1*. Similarly, for a given signal, the specificity of its derived path depends on the

combinations of SSCs and targets. Taking the signal *CDH3* as an example, *FOS* is in general the dominant SSC that mediates the regulation of target genes; many tumor-related genes, including the keratins and *S100* gene family, account for a substantial part of the most specific targets of *CDH3* (Supplementary Fig. 4h). Overall, the fidelity and specificity analysis helps to identify the most possible regulatory paths in the presence of crosstalk.

It is worth noting that, a high correlation between the fidelity and the specificity of a same pathway is not expected (Supplementary Fig. 4i). To give an example, we focus on the target gene *KRT17*, producing a long non-coding RNA that is a main regulator in cell cycle and tumor proliferation[61]. Its expression levels are affected by dozens of crosstalk pathways (Supplementary Fig. 4j). For example, *CDH3-RBBP5-KRT17*, has rather high specificity because *RBBP5* almost exclusively regulates the expression of *KRT17* rather than other targets; however, its fidelity is not necessarily high, i.e., *RBBP5* does not contribute much to the expression of *KRT17*, indicating that it is not a common transcription factor in cancer cells.

Finally, we ask whether the crosstalk pathways exert different patterns across different types of cells. To address this, we study the

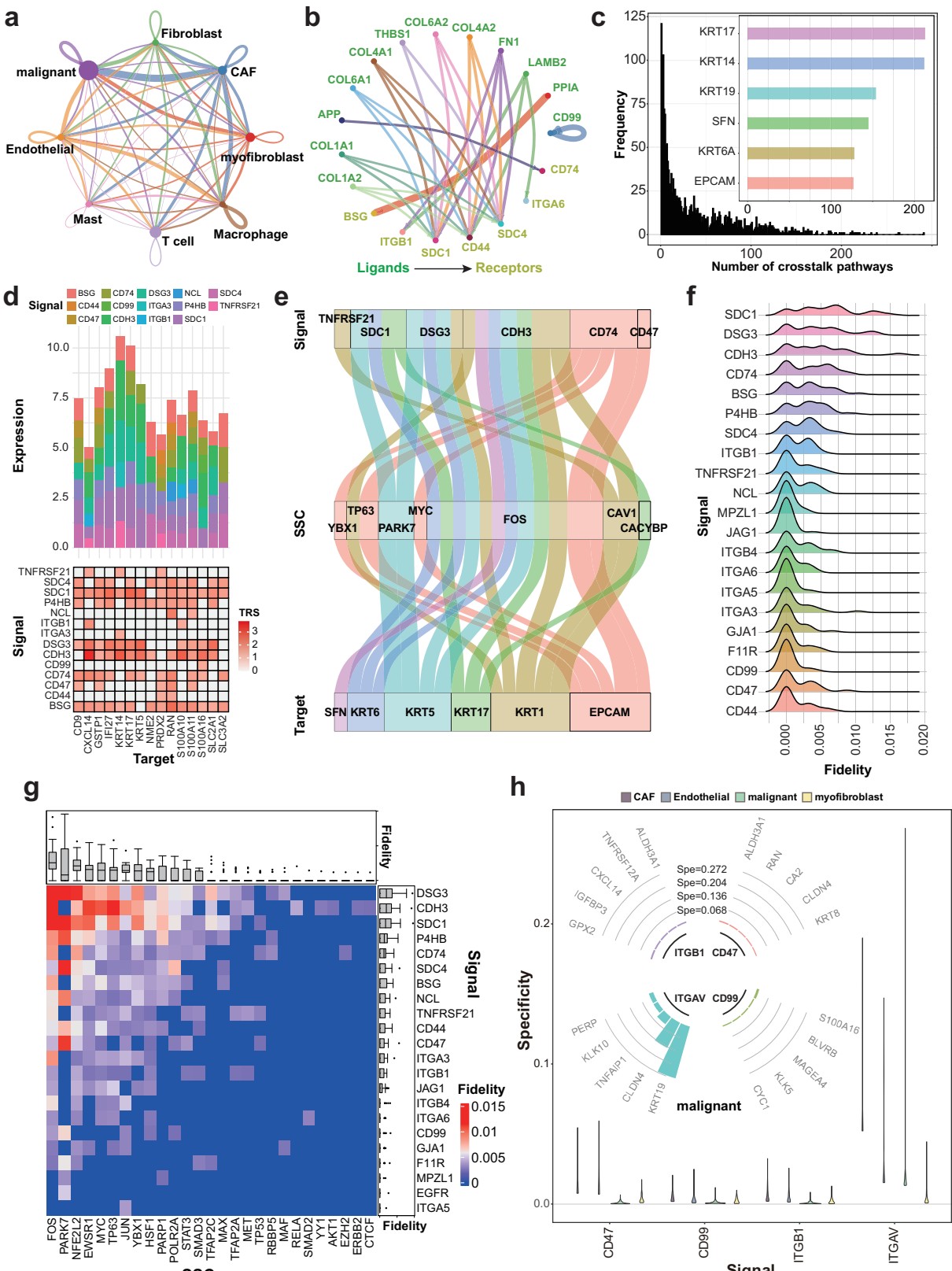

four main cell clusters of the HNSCC dataset: T cells, CAFs, myofi-broblasts and malignant cells. For the four signals shared by these four cell types, we calculate the specificity distributions of their derived paths. For three of the four signals (i.e., *CD47*, *CD99* and *ITGB1*), the malignant cells tend to have lower specificity, compared to non-malignant cells (Fig. 4h). This is consistent with the simultaneous aberrant activation of multiple signaling pathways in cancer cells[62]; the low specificity allows cancer cells to bypass inhibited pathways by using alternative routes, leading to drug resistance[63]. Interestingly, the *ITGAV-KRT19* path shows relatively high specificity, indicating a possible mechanism underlying tumor progression and metastasis, providing a potential intervention target for cancer therapy.

**Fig. 4 | Application of SigXTalk to the HNSCC dataset. a** The CCC patterns among cell clusters. The width of links represents the communication strength. **b.**The circle plot of the top-30 activated ligand-receptor pairs with the highest interaction strength with malignant as the receivers. The width of links represents the interaction strength. **c** The distribution of the number of crosstalk pathways for each target gene in malignant. Inner panel: the number of crosstalk pathways of 6 HNSCC markers. **d** The signal-target regulatory pattern of the top-15 targets that have the most crosstalk pathways in malignant. Upper: the contribution of the signals to the expression of these targets; Lower: the heatmap of the total regulatory strength (TRS) values for each signal-target pair. **e** The alluvial diagram of the top-20 pathways targeting the 6 HNSCC markers with the highest PRS values. The width of flow represents the PRS value. **f** The ridge diagram for the fidelity distribution of each SSC given the target *KRT5*. **g** The heatmap of the fidelity of each pathway that regulates the target *KRT17*. The boxplots illustrate the distribution of fidelity (Fid) values for each signal or SSC (sample size: 27 SSCs for each signal and 22 signals for each SSC). **h** The distribution of each target's specificity regulated by 4 common signals in 4 cell clusters. Inner panel: the top-5 targets regulated by 4 signals in the malignant cell cluster with highest specificity. Boxplot elements: center line, median; box limits, upper and lower quartiles; whiskers, 1.5x interquartile range; points, outliers. Source data are provided as a Source Data file.

## Contrasting regulatory patterns between healthy and diseased tissues

To study SigXTalk's ability of revealing the difference in intracellular regulatory patterns between different pathological conditions, we study the scRNA-seq datasets of lung cells from a severe COVID-19 patient (COVID) and a healthy control (HC)[64]. The data contains the expression of ~20,000 genes in 4636 and 5668 cells, respectively (Supplementary Fig. 5a, b). We focus on the CCC signals that regulate the expression of differentially expressed genes in fibroblasts, as pathological fibrosis is strongly correlated with the progress of lethal COVID-19. CCC inference analysis indicates similarities and differences in the major LR pairs that target fibroblasts between the COVID (Fig. 5a) and the HC sample (Fig. 5b). Specifically, Collagen and Laminin signals dominate the communication to fibroblasts. Communications among fibroblasts themselves account for a large part of the total CCC, while Endothelial and neuronal cells are the secondary senders (Supplementary Fig. 5c–f). Furthermore, based on CellChat, seven common signals are identified to be active in both samples while the others are activated exclusively in either HC (i.e., *UNCSC* and *EGFR*) or COVID (i.e., *ITGB1*, *ITGA1* and *ITGA2*) sample (Fig. 5c). We then ask whether these common signals can lead to similar or different intracellular regulatory patterns by using SigXTalk. To address this, we focus on two common signals *CD44* and *PTPRM*, whose downstream genes are highly activated in the fibroblasts of both samples, and first compare the number of the crosstalk pathways regulated by them between the two samples. *CD44* on average induces more downstream crosstalk pathways in the COVID sample (*p*-value of two-sided t-test: $< 2.2 \times 10^{-16}$, Fig. 5d). However, the overall PRS values in the COVID sample are significantly lower than that of the HC sample (*p*-value of two-sided t-test: $< 2.2 \times 10^{-16}$, Supplementary Fig. 5g), and the specificity of pathways activated by *CD44* in COVID sample are lower on average (*p*-value of two-sided t-test: $< 2.2 \times 10^{-16}$, Fig. 5e). This may relate to the severe lung fibrosis during COVID, leading to broad but non-specific activation of downstream pathways[65–67]. On the other hand, *PTPRM* exhibits overall higher number of activated crosstalk pathways and slightly higher PRS values (*p*-value of two-sided t-test: $7.4 \times 10^{-5}$) but lower specificity (*p*-value of two-sided t-test: $< 2.2 \times 10^{-16}$) in the COVID sample, compared to HC. We also investigate the fidelity of *CD44* and *PTPRM* for their target genes (Fig. 5f). The fidelity distribution of *CD44* shows little difference between COVID and HC samples, while *PTPRM* in the COVID sample exhibits much higher fidelity, meaning that the targets of *PTPRM* are mainly regulated by it rather than by other signals. Different from *CD44* and *PTPRM*, some signals' downstream genes are predicted to be exclusively activated in either of them (Fig. 5g). For example, *ITGB1* signaling is found to be activated in COVID sample by both CellChat and SigXTalk, with top 5 highest-specificity genes as *FN1*, *TGM2*, *LUM*, and *COL3A1* and *VCAN* (Supplementary Fig. 5). While CellChat predicts *ROBO1* and *ROBO2* signaling in both samples (Fig. 5a–c), but SigXTalk predicts they only activate the downstream genes in HC sample, providing finer granularity by distinguishing sample-specific downstream gene activation.

A gene could act simultaneously as multiple identities in the context of crosstalk. For example, *STAT3* is not only a key TF during the COVID-induced fibrosis but is also regulated by other signals or TFs. As

an SSC, we find that *PTPRM* is the dominant upstream signal that regulates the activity of STAT3 (Fig. 5h), by taking part in *STAT3*'s dephosphorylation[68,69]. Interestingly, the path *PTPRM-STAT3-RAB2A*, showing the highest pathway regulatory strength, remains underexplored. Considering the function of these individual genes in fibroblasts, the *PTPRM-STAT3-RAB2A* axis might contribute to excessive ECM deposition by promoting collagen synthesis (via STAT3) and secretion (via RAB2A), and thus contributing to remodeling of the lung ecosystem during the process of COVID[70,71]. This suggests a potential direct relationship between PTPRM's functionality and COVID-19, a molecular mechanism underlying the disease. On the other hand, the expression or activation of *STAT3* can be regulated by other SSCs, like *BACH2*, *JUN*, *ZEB1* and *MITF* (Fig. 5i). Among them, *ZEB1* is notably the key SSC that integrates three CCC signals with similar fidelity values. Such interplay between *ZEB1* and *STAT3* may contribute to the pathogenesis of pulmonary fibrosis induced by COVID-19[67,72].

## Tracking changes in crosstalk over time

Next, we study a scRNA-seq dataset with temporal snapshots to see if SigXTalk can track the crosstalk pathways, signal fidelity and specificity over time. In this dataset, the gene expression dynamics of the mouse lung cells at multiple time points after being injured by bleomycin is tracked[73] (Fig. 6a). Three combinations of time points are selected: Day 2 + Day 3, Day 10 + Day 11, and Day 15. After preprocessing (Supplementary Note 2), there are 4222, 3837 and 2058 cells for each time point, respectively (Supplementary Fig. 6a–c). We focus on Krt8+ ADI cells as the receiver cells, which characteristically express a rather high level of Keratin-8 after the bleomycin injury. Compared with the first 2–3 days after injury, Krt8+ ADI cells receive 3-4 times more CCC signals as the lung cells recover from the bleomycin injury (Fig. 6b, Supplementary Fig. 6d–f). In the meantime, the number of crosstalk pathways for each SSC has a similar growing tendency over time (Fig. 6c). In particular, transcription factors like *Jun* have plenty of pathways that pass through them for most of the time points, implying that they serve as critical hubs in the gene-gene regulation (Supplementary Fig. 6g–i). In terms of the fidelity, however, after the lung cells are injured, the overall fidelity of signals significantly decreases dramatically (Fig. 6d).

Furthermore, highly activated regulatory pathways are identified at different time points. At each time point, top-20 pathways with the highest PRS values are selected. It is found that *F11r* and *Cldn3* are the two main active signals that participate in the downstream regulatory pathways at all time points (Fig. 6e). Some signals, like *Sdc1*, are the main signals only after Day 10, implying that the pathways they regulate may participate in the regeneration of lung cells after injury. During the bleomycin injury and regeneration, *Ctnnb1* is one of the main SSCs of regulatory pathways at all time points. Its main targets include *Tpm1* and *Ccnd1*, which play important roles in accelerating or suppressing the proliferation of cells[74,75]. These observations imply that the up-regulation of these pathways may facilitate the recovery of lung cells but prevent them from being over proliferation.

Finally, we analyze the regulatory pathways that show a high level of heterogeneity across different time points using the target gene *Actb* as an example. We calculate the corresponding fidelity and

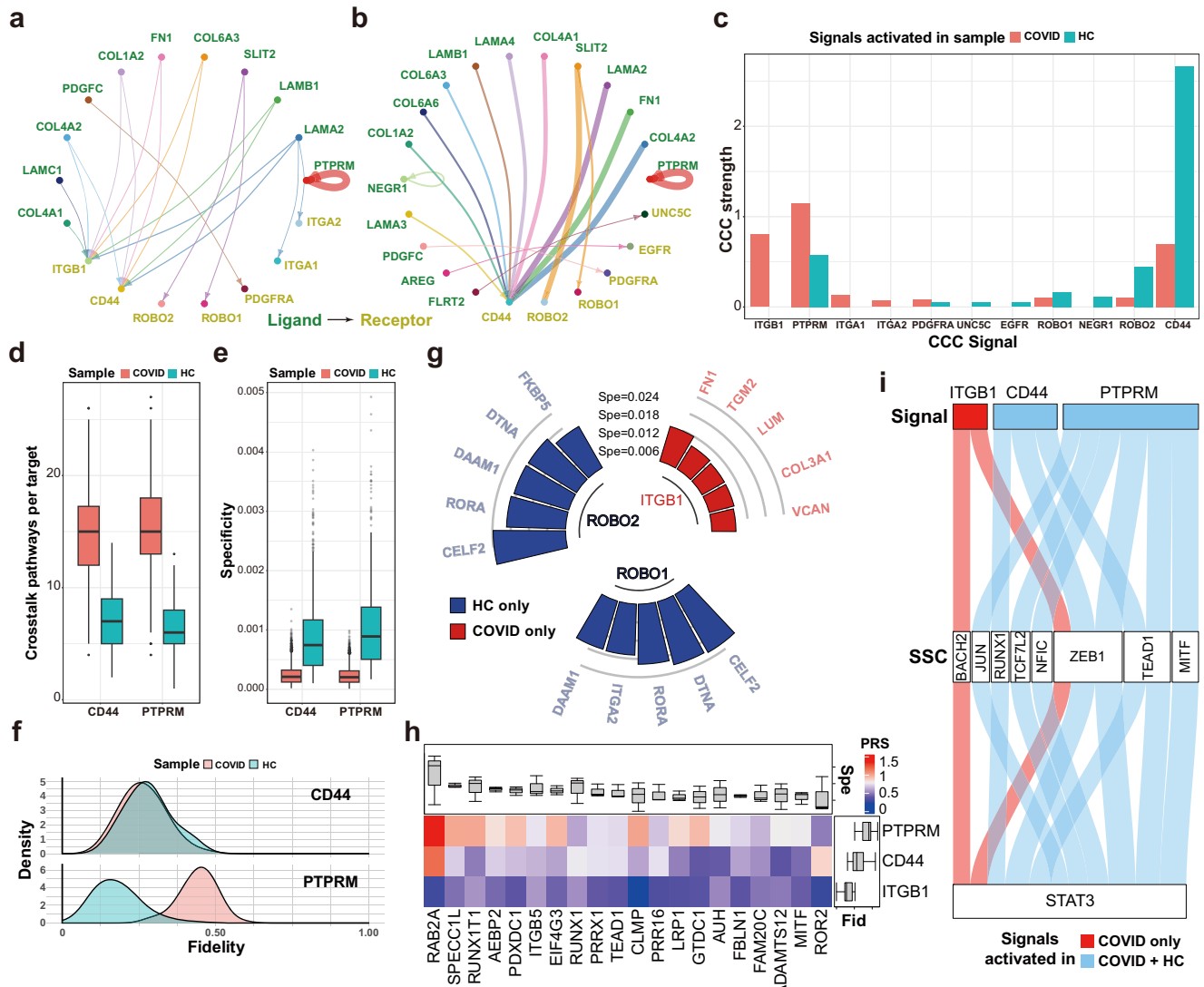

**Fig. 5 | Application of SigXTalk to the lung datasets of a COVID-19 patient (COVID) and a healthy individual (HC). a, b** The circle plot of signaling LR pairs that target fibroblasts in the COVID (**a**) and the HC (**b**) sample. The width of links represents strength of the LR pair. **c** The overall CCC strength of the receptors activated in the COVID and HC sample. **d**. The boxplot of the average number of crosstalk pathways regulating a target, regulated by *CD44* (left, sample size: 244 for COVID, 153 for HC) and *PTPRM* (right, sample size: 280 for COVID, 154 for HC). **e** The boxplot for the specificity distribution of targets, regulated by *CD44* (left, sample size: 4154 for COVID, 1140 for HC) and *PTPRM* (right, sample size: 4346 for COVID, 969 for HC). **f** The probability distribution of the fidelity measures of each target.

**g** The cyclized bar chart of the top-5 most specific targets regulated by sample-specific signals. **h** The heatmap of the PRS values of each pathway mediated by the SSC *STAT3* in the COVID sample. The upper and right boxplots annotate the specificity (Spe) of each target (sample size: 3 signals for each target) and the fidelity of each signal (sample size: 20 targets for each signal), respectively. **i** The alluvial diagram of the pathways regulating the target *STAT3* in the COVID sample. The width of flow represents PRS value of the corresponding pathway. Boxplot elements: center line, median; box limits, upper and lower quartiles; whiskers, 1.5x interquartile range; points, outliers. Source data are provided as a Source Data file.

specificity of the pathways regulating this gene, showing the various regulation patterns (Fig. 6f). *F11r* and *Cdh1* are found to be the only two common signals that regulate *Actb* at all the time points. The fidelity values of corresponding regulatory pathways are highly dependent on their SSCs. For example, only the *Cdh1-Ctnnb1-Actb* (on Day 2 + Day 3) and *F11r-Eef1a1-Actb* (on Day 15) pathways remain relatively high fidelity among them. Instead, *F11r* regulates *Actb* via different SSCs with high specificity, i.e., *Mtdh* on Day 2 + Day 3, *Eef1a1* on Day 10 + Day 11 and Day 15.

## Refining cell-cell communication inference by incorporating crosstalk

Often, only part of the cells in the receiver's cluster participate in the CCC. Target genes regulated by the receptor in the interacted cells express differentially compared with not-interacted ones. To

investigate how SigXTalk can refine CCC using the information from downstream regulatory pathways, we study a mouse brain 10X Visium dataset which contains spatially resolved transcriptomic data (Supplementary Fig. 7a). In total, nine main clusters are identified. CCC analysis based on the spatial coordinates of cells further indicates that the sender and receiver cells interact only if they are close enough (Fig. 7a). Specifically, the results suggest strong CCC from Meis2 cells to L6 IT cells, whose activated receptors include Syndecan (*Sdc*) genes, *Oprl1*, *Lrp1*, *Npy1r*, *Ntrk3*, *Ptprz1*, and *Ncl* genes (Fig. 7b). For these receptors (as signals), we first identify the pathways that have high specificity values. Specifically, the top-5 targets along with their corresponding SSCs are selected and visualized for each signal (Fig. 7c). Several target genes, like *Nrgn*, *Camk2n1*, *Calm1* and *Calm2* are highly specific regardless of the upstream CCC signals, indicating that they have very high transcription activity and priority in LR IT cells. These

targets could become clues for identifying the cells that do participate in CCC. Specifically, we consider which cells receive CCC signals from Meis2 cells via the *Ntrk3* receptor. Using the highly specific targets, all the L6 IT cells are subclustered with a weighted *k*-nearest neighbors

(kNN) algorithm (see Methods for details). Intuitively, the L6 IT cells that indeed receive the communication signals from Meis2 cells are closer to the senders than not-interacted ones (Fig. 7d). This observation is further supported by calculating the distances of interacted

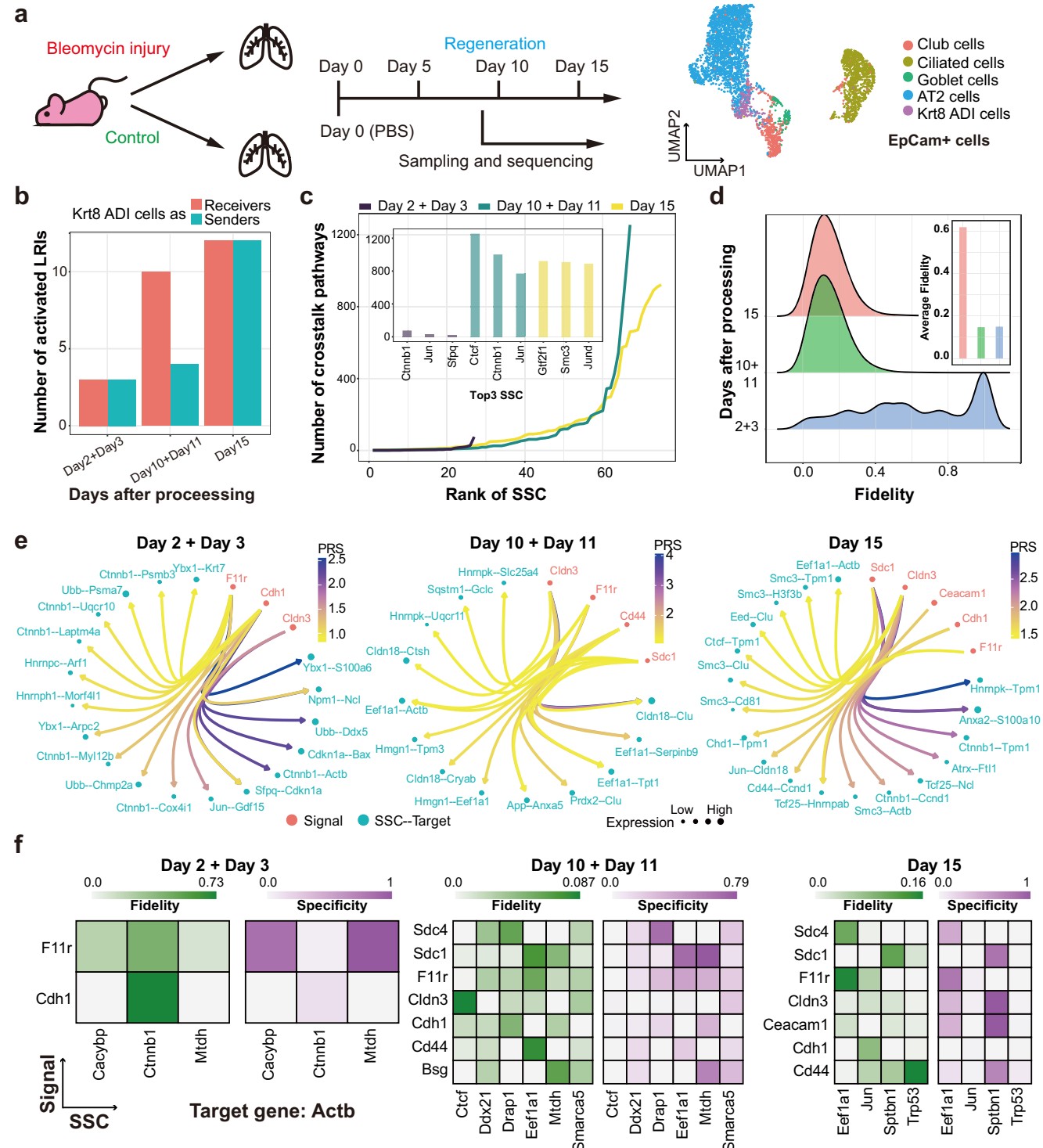

**Fig. 6 | Application of SigXTalk to the time-resolved mouse lung dataset. a.** The experimental process of bleomycin injury on mouse lung cells. Cells are visualized using Uniform Manifold Approximation and Projection (UMAP); **b.** The bar chart that visualizes the number of activated ligand-receptor interactions (LRIs) regarding Krt8 ADI cells at different time points; **c.** The number of crosstalk pathways that pass through the SSCs at each time point. The SSCs within each time point are ranked according to the numbers of crosstalk pathways. The top-3 SSCs with most crosstalk pathways are highlighted in the inner panel; **d.** The probability

distribution of all the fidelity values at each time point. The average fidelity at each time point is highlighted in the inner panel; **e.** The circle plots of the top-20 pathways with the highest PRS values at each time point; **f.** The heatmaps of the fidelity and specificity of crosstalk pathways regulating *Actb* at different time points. Here, the fidelity or specificity is calculated based on the crosstalk module containing pathways with common SSC + Target or Signal + SSC, respectively. Source data are provided as a Source Data file.

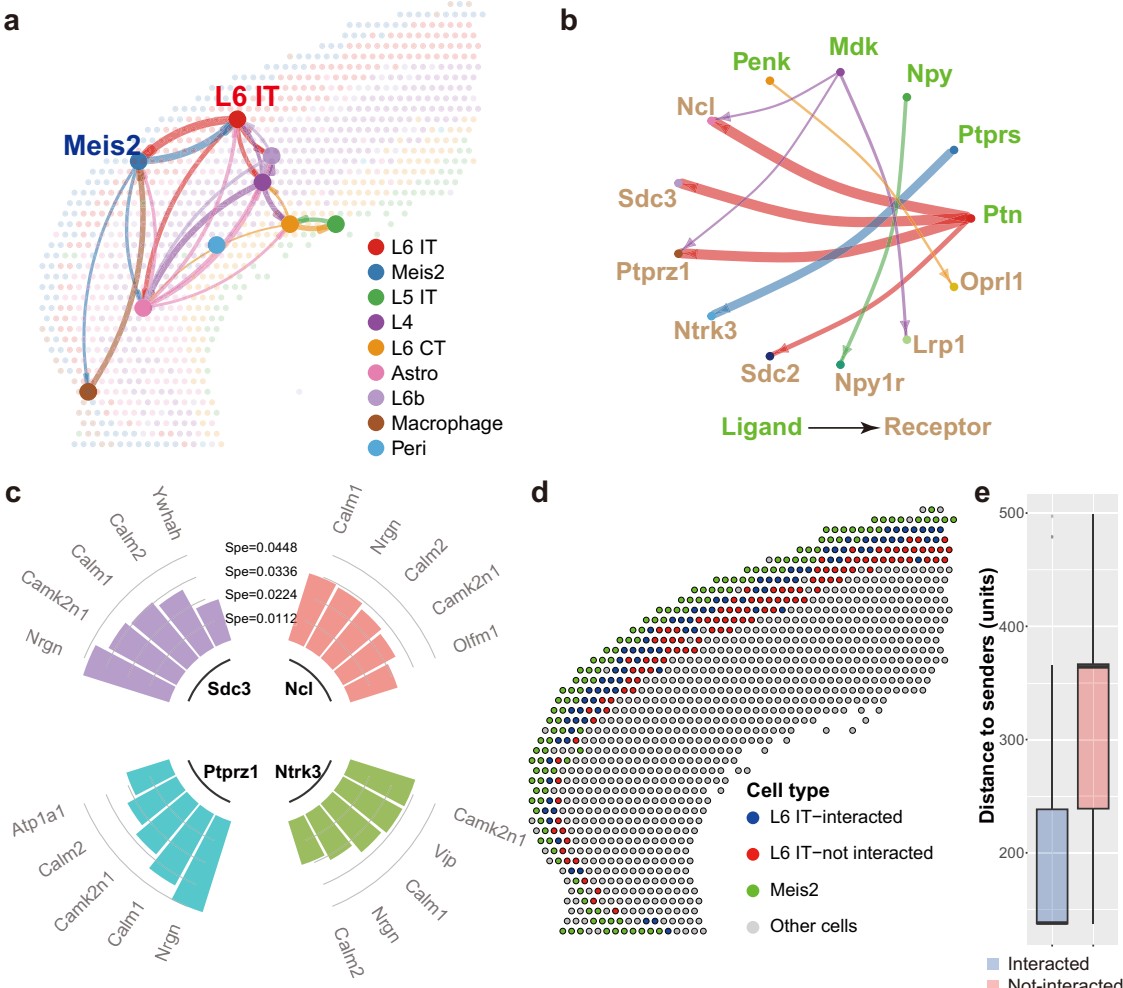

**Fig. 7 | Application of SigXTalk to the spatially resolved mouse brain dataset. a.** The spatial diagram of the CCC strength between cell groups. The node that represents each cell cluster is located on the geometric center of the cells within the cluster. The width of each link represents the pairwise CCC strength; **b.** The circle plot of signaling LR pairs that target L6 IT cells; **c.** The cyclized histograms of the top-5 target genes regulated by CCC signals with the highest specificity (Spe); **d.** The spatial plot of the L6 IT cells subclustered by the interactions with Meis2 cells; **e.** The box plot of the distance measures from L6 IT cells to the Meis2 cells based on the subclustering results (sample size: 88 interacted spots and 108 not-interacted spots). Boxplot elements: center line, median; box limits, upper and lower quartiles; whiskers, 1.5x interquartile range; points, outliers. Source data are provided as a Source Data file.

vs. not-interacted receiver cells to sender cells. Specifically, the median distance between Meis2 cells and interacted L6 IT cells is 62% shorter than that of not-interacted L6 IT cells (p-value of two-sided t-test: $< 2.2 \times 10^{-16}$, Fig. 7e). Additionally, the cells that participate in CCC via other activated CCC signals including *Ncl*, *Sdc3* and *Ptprz1* are identified. Again, the interacted cells predicted by these signals are closer to sender cells (Supplementary Fig. 7b–d). In summary, SigXTalk can distinguish the cells according to their response to CCC signals, which leads to a finer CCC landscape.

## Discussion

SigXTalk is designed to identify, quantify and visualize the crosstalk induced by CCC events. Unlike existing methods that infer pairwise (ligand-receptor, TF-target or ligand-target) regulatory interactions, we are focused on the role of genes that different regulatory pathways share as well as the effect of crosstalk on the target genes. With the introduced concepts of fidelity and specificity, the pathways with shared genes are measured and compared in multiple ways with different focuses. By applying SigXTalk to multiple datasets, we show that crosstalk is a universal regulatory mechanism across different cells, tissues and species. Our findings also indicate that in different cells, the

expression of the same target gene may be affected by different regulatory pathways with different levels of fidelity or specificity. With the growing number of single-cell transcriptomic datasets with different biological conditions or at different time points, SigXTalk can reveal the difference or evolution among samples, providing new ways in contrasting regulatory landscape between biological conditions. With more detailed downstream gene regulations, SigXTalk can be used to refine CCC inferences between senders and their specific receivers, by identifying the cells whose target genes are indeed activated by the signals from the senders.

Graph or network structures, based on pairwise interactions between nodes (variables), have been widely used in scRNA-seq data analysis[9,76–79]. However, little attention has been paid to the higher-order structure existing between genes and cells[80,81]. SigXTalk addresses this gap by leveraging a hypergraph framework to mine such higher-order regulatory relationships among genes. Compared with SigXTalk, the other GRN inference methods do not perform well when the actual interaction from the regulator to the target has intermediate regulators by other molecules. Because the evaluation metric is designed for capturing the shared common regulators, the other general GRN inference methods naturally underperform using the

such a metric. This caveat should be considered when interpreting the comparative results. Meanwhile, methods that directly measure the regulatory effect from the regulator to the target tend to overlook the specific pathways along which the regulatory signal transduces. SigX-Talk fills the gap between these two types of methods, as the higher-order regulatory relationship is characterized using hypergraphs.

The regulatory relationship in cells could be directional − and so do the edges in graphs and hyperedges in hypergraphs. The direction information of regulatory relationship may be connected with the causal information flows among genes. Recently, causality studies have been carried out for various biological systems[82], including ecosystems[83], metabolic pathways[84] and GRNs[85]. Causal inference algorithms, like Granger causality[86], cross mapping[83,87], structural equation modeling (SEM)[88] have been developed for different complex systems from stochastic to deterministic ones. However, its application to scRNA-seq data analysis has not been fully explored, with few studies[47,89] restricted to pairwise causal relationships. As hypergraphs naturally capture the beyond-pairwise interactions among variables, SigXTalk can be further improved towards a more accurate prediction of causal relationship between genes, by incorporating the causal inference algorithms to construct the prior hypergraph, or to estimate the PRS values of a pathway from the perspective of biologically related causality.

Related to pathway inference and causality, here, we focus on two scenarios regarding cell's responses to input signals where SigXTalk could be adopted. First, as illustrated in our second example, the activity of crosstalk pathways may vary across different pathological conditions. In this context, the condition of a cell (or a group of cells) is regarded as a label, so that the key regulatory pathways can be identified using supervised machine learning techniques. For instance, PathFinder[90] employs a graph transformer trained on the scRNA-seq data from cells of different conditions (e.g., diseased vs healthy). Its output includes the path score of the Ligand-Receptor-Mediator-Target pathway, which measures the relationship between this pathway and the diseased/healthy condition of the cells. The hypergraph learning framework in SigXTalk could be modified to estimate such importance of a pathway in shaping different conditions. Furthermore, SigXTalk could be modified for predicting the response to the knockout or knockdown of CCC signaling genes (ligands or receptors). So far, methods like scTenifoldKnk[91], CellOracle[92] and GenKI[93] have been applied to address similar problems by constructing the intracellular GRN and simulating the response of perturbation of regulator genes. We point out that SigXTalk could be potentially used as an alternative tool for this task, as the concepts of fidelity and specificity naturally correspond to a target's response to CCC signals, especially in the presence of crosstalk. Availability of ligand/receptor knockout experiments will allow us to carry out more comprehensive analysis and test the predictions made by SigXTalk.

Several limitations of SigXTalk are worth mentioning. First, the pathway regulatory strength is calculated using the static expression profile of genes. However, transcriptomic regulation is a dynamic process, thus the fidelity/specificity analysis could be further improved if the time-course data is available. For instance, methods like SCODE[45], GRISLI[25] and spliceJAC[94] incorporate the information from RNA splicing and reconstruct the dynamical system of gene expression. SigX-Talk may adopt similar approaches, for example, by fitting linear differential equations that the expression of genes obeys and extracting the coefficients of the linear terms as the strength of gene-gene interactions. Second, SigXTalk considers only the intracellular regulatory pathways that contain receptors (signal), TFs (SSC) and target genes (target). This simplification may neglect the crosstalk effect at other levels, for example, the intermediate molecules from receptors to TFs. Theoretically, the hypergraph learning framework in SigXTalk is applicable for non-uniform hypergraphs (i.e., the lengths of hyperedges in the hypergraph are not necessarily the same), allowing

us to identify the activated pathways that contain an arbitrary number of genes. Such an extension will allow SigXTalk to be further improved and generalized. For example, a hyperedge may contain all the genes along the pathway, including the signal, SSC, target and any other mediators from the signal to the SSC. In this case, the PA value of the hyperedge can be described as a linear combination of concatenated low-dimensional representations of nodes along this hyperedge. Third, in many cases, the interactions between genes are bidirectional, forming positive or negative feedback loops, a complex situation beyond this study. Using dynamical models with feedbacks, in principle, can improve estimation of pathway regulatory strength. Finally, like many other computational methods for regulatory network inference (e.g., SCENIC), our method may filter out lowly expressed TFs but with critical regulatory roles, because low-expression TFs may represent unreliable or noisy data. High-sensitivity sequencing methods, such as single-molecule RNA or targeted RNA sequencing, could improve the detection of rare transcripts[95,96]. Imputation techniques to enhance the signal of sparsely expressed genes or applying context-specific models that consider the functional impact of TFs regardless of expression levels might better capture the roles of these TFs[97].

Single-cell multiomics data, like paired scRNA-seq and scATAC-seq (Assay for Transposase-Accessible Chromatin with high through-put sequencing)[98], can provide additional important regulatory information. Recent studies have found that incorporation of scATAC-seq data may help in identifying activated TF-TG pairs and constructing more accurate and robust gene regulatory networks[77,99,100]. With such incorporation, the accuracy and efficiency of fidelity/specificity calculation could be improved, as the activation information for TF-TG pairs could be used while training the hypergraph neural networks. This will also help to indirectly infer the activity of lowly expressed TFs. Moreover, spatial transcriptomics has been widely used to facilitate a more effective and reliable CCC inference[9,11]. SigXTalk could benefit from spatially resolved RNA expression data in predicting the effect of CCC on the expression of downstream targets more accurately.

In summary, by identifying pathways with shared molecules and quantifying their regulatory selectivity in terms of fidelity and specificity, SigXTalk takes a crucial step towards a more comprehensive cognition of the regulatory landscape composed of intertwined pathways from CCC signals to target genes.

## Methods

### Data input and preprocessing

SigXTalk takes the single-cell RNA expression matrix as the input. And pre-assigned labels are required for the dataset to cluster the cells. Rare cell clusters, which are defined as clusters containing less than 20 cells, are filtered out. For the five real-world datasets, quality control is performed to filter out the cells with abnormally high gene expression levels and high percentage of mitochondrial genes (Supplementary Note 2). Besides, we filter out the genes that are expressed in less than 5% of all the cells. Then the data is normalized and scaled following a standard workflow with the *NormalizeData* and *ScaleData* function in the Seurat R package[101]. The processed expression data could finally be represented as a matrix $\mathbf{X} = \{x_{ij}\}$, where $i = 1, 2, ..., N$ represents the genes and $j = 1, 2, ..., M$ represents the cells.

### Inference of cell-cell communication (CCC)

Given the normalized expression matrix, we begin with inferring the cell-cell communication using the CellChat software[13]. CellChat first calculates the ensemble average expression levels of ligands in the sender cell cluster and their corresponding receptors in the receiver cell cluster based on a built-in ligand-receptor interaction (LRI) database. Then, a Hill function based on the law of mass action is adopted to measure the strength of the communication between a given LRI[102]. At the same time, the expression of auxiliary molecules like agonists and antagonists are also incorporated to improve the accuracy of CCC

inference. Given the sender cluster $i$ and receiver cluster $j$, the output of the CellChat contains the communication strength of the $k$-th LR pair (denoted as $P_{ij}^k$) and its significance level (denoted as the $p$-value pv). Specifically, the communication strength is calculated as:

$$P_{ij}^k = \frac{L_i R_j}{K_h + L_i R_j} \times \left(1 + \frac{AG_i}{K_h + AG_i}\right) \times \left(1 + \frac{AG_j}{K_h + AG_j}\right) \times \frac{K_h}{K_h + AN_i} \times \frac{K_h}{K_h + AN_j} \times \frac{n_i n_j}{n^2}. \quad (1)$$

Here, $L_i$ and $R_j$ represents the expression levels of the ligand and receptor in the cell cluster $i$ and $j$, respectively. If the ligand or the receptor is composed of multiple subunits, then $L_i$ or $R_j$ is measured as the geometric mean of the expression levels of the corresponding subunits. $AG$ and $AN$ respectively denote the average expression of agonists and antagonists from both sender and receiver cells. $n_i$, $n_j$ and $n$ represent the population of sender cells, receiver cells and all cells, respectively. $K_h$ is the parameter of the Hill function which is set to 0.5 by default.

The significance level of an LR pair is measured by a permutation test by randomly permuting the group labels of cells, and then recalculating the communication probability $P_{ij}^k$ between cell group $i$ and cell group $j$ through this LR pair. The $p$-value is specified as:

$$\text{pv} = \frac{\{\#m | P_{ij}^{k,m} \geq P_{ij}^k, m = 1, 2, \dots, M\}}{M}, \quad (2)$$

where $P_{ij}^k$ denotes the communication probability of the $m$-th permutation ($M$ permutations in total). The LRIs with $p$-value > 0.1 are filtered out. The total communication strength of a receptor is calculated by summing the strength of all the remaining LRIs that targeting this receptor. Finally, receptors whose total communication strength is higher than 10% of the maximum of them are selected and recognized as signal nodes in the regulatory hypergraph.

## Prior databases of intracellular gene-gene interactions
We collected and processed the gene-gene interaction data from NicheNet[28] to form the prior databases of intracellular regulatory pathways. NicheNet database was chosen for its integration of gene interactions from multiple sources, providing broader coverage and a more comprehensive view of signaling pathways compared to single-source databases. Specifically, the databases could be divided into the database that contains interactions between receptors and their downstream transcription factors (TFs), and the database that contains the TFs and their downstream target genes.

## Construction of intracellular regulatory hypergraph
For the intracellular regulatory system that begins with receptors and ends up with target genes, TFs could be regarded as transit hubs that receive CCC signals from multiple receptors and regulates their downstream target genes accordingly. In the context of crosstalk pathways, therefore, receptors, TFs and target genes could be identified as signals, shared signaling components (SSCs) and targets. Such diverse identities and the regulatory relationships among them form the basis of hypergraph construction and the representative learning on it.

Hypergraphs, which measure the interactions among more than two objects, could be viewed as a conceptual generalization of common graphs (networks). Formally, a hypergraph is represented by a pair $\mathcal{H} = (V, \mathcal{E})$, where $V = \{v_p\}, p = 1, 2, \dots, N_v$ is the set of nodes and $\mathcal{E} = \{e_q = (v_{q1}, v_{q2}, v_{q3}, \dots)\}, q = 1, 2, \dots, N_e$ is the set of hyperedges (see Supplementary Note 3 for details). In this article, the nodes represent all the genes in the dataset as well as in the databases and the (undirected) hyperedges represent the regulatory relationships among them. To build a hypergraph that describes the regulatory pathways from signals to targets, SigXTalk first computes the interaction measures among different types of nodes. Specifically, a possible regulatory pathway from the signal to the target could be represented by a combination of nodes $(v_{si}, v_{cj}, v_{tk})$, where the subscript $s$, $c$, and $t$ refers

to signal, SSC and target, respectively. And the initial probability of them forming a hyperedge is:

$$S_H\left(v_{si}, v_{cj}, v_{tk}\right) = \left| \text{PPR}\left(\mathbf{x}_{si}, \mathbf{x}_{cj}\right) \times \text{Cor}\left(\mathbf{x}_{cj}, \mathbf{x}_{tk}\right) \right|, \quad (3)$$

where $\text{PPR}(\mathbf{x}, \mathbf{y})$ denotes the PageRank score from signal $x$ to SSC $y$ with the bold letter representing the vector of expression. Given the signal-SSC (Receptor-TF) dataset represented by a graph, the signal $x$ is set as the seed and $\text{PPR}(\mathbf{x}, \mathbf{y})$ measures how much gene $x$ and $y$ are connected via all the possible regulatory links in the dataset[55]. $\text{Cor}(\mathbf{y}, \mathbf{z})$ denotes the Spearman's correlation coefficient between the scaled expression levels of gene $y$ and $z$ in the receiver cell cluster. Considering the high sparsity of real-world gene regulatory networks, a threshold $\varphi$ is set to filter highly correlated combinations and binarize the hypergraph:

$$A_H\left(v_{si}, v_{cj}, v_{tk}\right) = \begin{cases} 1 & , \text{if } S_H\left(v_{si}, v_{cj}, v_{tk}\right) > \varphi, \\ 0 & , \text{otherwise}. \end{cases} \quad (4)$$

Consequently, the hyperedges in the constructed hypergraph are exactly the node combinations with $A_H = 1$. Here, the threshold $\varphi$ is selected so that only the top-10% of the combinations with the highest $S_H$ are used to form the hyperedges.

## Identifying activated regulatory pathways via a hypergraph learning framework
Quantifying the regulatory strength of a pathway first requires an accurate identification of activated regulatory pathways from signals to targets, which could be viewed as the hyperedge prediction task. It is a natural extension to the classical edge prediction problem on common graphs and aims at predicting the missing hyperedges on a hypergraph using the observed hypergraph structure and its node features. To address this, we design a hypergraph deep learning framework to generate the low-dimensional embeddings of nodes and thereby estimate the probability of a node combination forming a hyperedge. This framework consists of two consecutive hypergraph neural network (HGNN) modules, followed by multilayer perceptrons (MLPs) with multiple channels. For the HGNN modules, we adopt a hypergraph convolution layer called HGNNConv + [103]. HGNNConv+ utilizes a spatial-based hypergraph convolution approach to pass messages among nodes via hyperedges and update the representation of a target node by aggregating features of its neighbors. Such update is achieved by (1) obtaining the feature of each hyperedge through a weighted sum of the node features that the hyperedge contains; (2) updating the node features through a weighted sum of the hyperedge features that contain the node. Mathematically, denoting $\mathbf{Z}_k$ as the representations of nodes in the $k$-th HGNNConv+ layer ($\mathbf{Z}_1$ simply represents the input expression matrix for convenience), the update process could be described as:

$$\mathbf{Z}^{k+1} = \sigma\left(\mathbf{D}_v^{-1} \mathbf{H} \mathbf{D}_e \mathbf{H}^\top \mathbf{Z}^k \mathbf{\Theta}^k\right). \quad (5)$$

In this equation, $\mathbf{H}$ represents the incidence matrix of the hypergraph, and $\mathbf{D}_v$ and $\mathbf{D}_e$ are the diagonal matrices of node degrees (the number of hyperedges that contain each node) and hyperedge degrees (the number of nodes that each hyperedge), respectively. $\mathbf{\Theta}^k \in \mathbb{R}^{C^k \times C^{k+1}}$ is a trainable parameter of the $k$-th layer, with $C^k$ denoting the dimension of the $k$-th node features. $\sigma$ represents the Exponential Linear Units (ELU) activation function. A detailed description of hypergraph concepts and HGNNConv+ is presented in Supplementary Note 3.

The output of the HGNN modules is the low-dimensional representations of the nodes. To predict the hyperedges among them, the

representations of the signal, SSC and target nodes are further encoded using two-layer MLPs respectively. The outputs of the MLP layers for each type of nodes are written as follows:

$$
\begin{aligned}
\mathbf{y}_{si} &= \sigma'\left(\mathbf{W}_2^\top \sigma\left(\mathbf{W}_1^\top \mathbf{z}_{si} + \mathbf{b}_1\right) + \mathbf{b}_2\right); \\
\mathbf{y}_{cj} &= \sigma'\left(\mathbf{W}_2^\top \sigma\left(\mathbf{W}_1^\top \mathbf{z}_{cj} + \mathbf{b}_1\right) + \mathbf{b}_2\right); \\
\mathbf{y}_{tk} &= \sigma'\left(\mathbf{W}_2^\top \sigma\left(\mathbf{W}_1^\top \mathbf{z}_{tk} + \mathbf{b}_1\right) + \mathbf{b}_2\right),
\end{aligned}
\tag{6}
$$

where $\mathbf{W}_1, \mathbf{W}_2$ denote the weight matrices of the linear neural network layers, $\mathbf{b}_1, \mathbf{b}_2$ denote the biases, $\sigma'$ represents the Rectified Linear Units (ReLU) activation function. Finally, for a given combination $(v_{si}, v_{cj}, v_{tk})$, the probability that they form a hyperedge is determined by the pathway activity (PA):

$$
\mathrm{PA}\left(v_{si}, v_{cj}, v_{tk}\right) = \left|\mathrm{Cos}\left(\mathbf{y}_{si}, \mathbf{y}_{cj}\right) \times \mathrm{Cos}\left(\mathbf{y}_{cj}, \mathbf{y}_{tk}\right)\right|,
\tag{7}
$$

where $\mathrm{Cos}(\mathbf{x}, \mathbf{y})$ denotes the cosine similarity between variable $\mathbf{x}$ and $\mathbf{y}$.

### Training of the hypergraph neural network

Hyperedge prediction in the SigXTalk framework could be performed in either supervised or self-supervised approaches. When the ground truth of regulatory pathways is accessible, the hypergraph model could be easily trained by splitting the ground truth into training and validation sets. However, like edge prediction on graphs, hyperedge prediction on hypergraphs also faces the difficulty of obtaining or generating appropriate positive and negative training samples. In particular, neither the ground truth hypergraph nor the training samples is easy to obtain in our task since the real gene-gene interactions are seldom known. To tackle this problem, we use a self-supervised training strategy by filtering the combinations with the highest probabilities when the ground truth is not available. Here the top-10% of the combinations with the highest $S_H$ is selected to make up the positive samples labelled as "1". The number of these positive samples is denoted by $N_{\mathrm{pos}}$. The negative samples (labelled as "0") of the same size are then generated and contains combinations with low $S_H$ and combinations not in the prior database. Both training and validation samples are randomly selected from the positive and negative samples and are set to contain equal numbers of positive and negative samples.

With the above notations, the loss function to be optimized in SigXTalk is the binary cross entropy (BCE) loss, which is given by:

$$
\mathrm{BCEloss} = -\sum_{m}^{N_m}\left(P_m \log(\mathrm{PA}_m) + (1 - P_m)\log(1 - \mathrm{PA}_m)\right),
\tag{8}
$$

where $P_m$ and $\mathrm{PA}_m$ represent the label and the estimated probability of the $m$-th sample, respectively, $N_m$ is the total number of samples in a training batch. An Adam optimizer[104] is used to train the neural network with an exponential scheduler to control the learning rate. More details of the training process are provided in Supplementary Note 4 and Supplementary Table 2.

With the well-trained neural network, the PA value of a pathway (a combination of signal, SSC and target nodes that occurs in the dataset) can be predicted. This value reflects whether regulatory pathway is truly activated by CCC in the receiver cells. We select the pathways with the threshold PA > 0.75 and name them as activated pathways.

### Measuring the pathway fidelity and specificity

The SSC and the target of an activated pathway together forms an SSC-target pair, whose Activation Strength (ACS) in the $m$-th cell could be measured by the product of their normalized expression levels:

$$
\mathrm{ACS}_m\left(v_{cj}, v_{tk}\right) = x_{cj,m} \times x_{tk,m}.
\tag{9}
$$

The ACS is then modelled using the signals of all the activated pathways that have the same SSC-target pair:

$$
\mathrm{ACS}_m\left(v_{cj}, v_{tk}\right) = f\left(x_{s1}, \ldots, x_{sl}\right).
\tag{10}
$$

The model is estimated using Random Forest, a tree-based statistical learning method that has been proved efficient in regression and classification tasks[40]. After fitting the above model using the gene expression data, the permutation importance score of each variable (signal) is extracted and regarded as the pathway regulatory strength (PRS) of the corresponding signal-SSC-target pathway. The permutation importance score of a feature is measured by randomly shuffling the values of this feature and re-evaluating the model performance. The decrease in the performance metric (relative to the baseline) due to the permutation of the feature is recorded as this feature's permutation importance.

If two or more pathways have the same signal, SSC (TF) or target, they are called crosstalk pathways and form a crosstalk module (Xmod). Generally, there are three types of crosstalk modules depending on the single component pathways have in common, namely the Signal-Xmod ($\mathrm{XM}_s$), SSC-Xmod ($\mathrm{XM}_c$), and the Target-Xmod ($\mathrm{XM}_t$). Given a pathway $e_q = (v_{si}, v_{cj}, v_{tk})$ within a Target-Xmod $\mathrm{XM}_t(v_{tk})$ (i.e. all the pathways that end up with target $v_{tk}$), its fidelity is defined as follows:

$$
\mathrm{Fid}\left(e_q\right) = \frac{\mathrm{PRS}(e_q)}{\sum_{e_r \in \mathrm{XM}_t(v_{tk})} \mathrm{PRS}(e_r)},
\tag{11}
$$

where PRS is calculated using the well-trained neural network. Similarly, for $e_q$ within the Signal-Xmod $\mathrm{XM}_t(v_{si})$, its specificity is defined as follows:

$$
\mathrm{Spe}\left(e_q\right) = \frac{\mathrm{PRS}(e_q)}{\sum_{e_r \in \mathrm{XM}_s(v_{si})} \mathrm{PRS}(e_r)}.
\tag{12}
$$

Intuitively, the fidelity of pathways describes the relative contributions of different CCC signals to a given target, while the specificity describes the 'regulatory priority' of a given CCC signal to different downstream targets.

Additionally, we can define a "compound" crosstalk module that contains pathways that share two components (Signal + SSC, or SSC + Target) simultaneously. In this case, the specificity or fidelity of a pathway may be similarly defined.

Furthermore, we can measure the total fidelity and specificity of a certain signal-target pair over all the possible SSCs. We first define the total regulatory strength (TRS) from signal $v_{si}$ to target $v_{tk}$ as:

$$
\mathrm{TRS}(v_{si}, v_{tk}) = \sum_{j=1}^{J} \mathrm{PRS}(v_{si}, v_{c1}, v_{tk}).
\tag{13}
$$

The total fidelity and specificity could be written as:

$$
\begin{aligned}
\mathrm{Fid}(v_{si}, v_{tk}) &= \frac{\mathrm{TRS}(v_{si}, v_{tk})}{\sum_r \mathrm{TRS}(v_{sr}, v_{tk})}, \\
\mathrm{Spe}(v_{si}, v_{tk}) &= \frac{\mathrm{TRS}(v_{si}, v_{tk})}{\sum_r \mathrm{TRS}(v_{si}, v_{tr})}.
\end{aligned}
\tag{14}
$$

SigXTalk allows a flexible quantification of fidelity and specificity according to the biological context without generosity.

### Method evaluation using simulated datasets

We first evaluate the performance of SigXTalk and other representative methods using simulated data produced by SERGIO[41]. To simulate the signal transduction from receptors to target genes, we generate a 3-layer regulatory network, including the random gene-gene

interaction between receptors and TFs, and between TFs and target genes. Specifically, for a preset number of genes (200 genes by default), we split them into receptors, TFs and target genes, with the ratio 1:4:15. The gene-gene interaction between receptors and TFs is constructed by randomly connecting them with a fixed probability, which measures the density of the receptor-TF network. The TF-target gene network is constructed similarly. The regulatory strength of each pair of genes is randomly generated following a uniform distribution. Finally, the receptor-TF network and the TF-target gene network are merged to form a large regulatory network as the input of SERGIO.

Given the above regulatory network, SERGIO employs the chemical Langevin equation (CLE) to simulate the genes' expression levels, where the gene regulatory relationship is reflected by the production rate of the gene to be regulated. Furthermore, technical noise is incorporated into the synthetic data to simulate the real scRNA-seq data more realistically. The default number of cells is set to 1500, which have previously been equally divided into three clusters.

To evaluate the performance of methods in recovering the activated regulatory pathways, the pre-defined receptor-TF-target gene pathways are recognized as positive samples, while the negative samples are randomly generated using the combinations of receptors, TFs and target genes that do not exist in the pre-defined pathways. Among all the samples, 80% of them are used as training samples while the remaining samples are used as test samples. All the methods produce a list that contains the probability that a combination of signals, TFs and targets form an activated pathway. Here, SigXTalk uses the PA value for this probability, while for other methods the probability is calculated by multiplying the absolute value of their signal-TF and TF-target scores. The performance of methods is measured by two evaluation metrics: the Area Under Receiver Operating Characteristic (AUROC) and the Area Under Precision-Recall Curve (AUPRC) using the *roc_auc_score* and the *average_precision_score* functions in the *sklearn* Python package, respectively[105]. The above simulation and evaluation are performed 10 times, with each time randomly generating the regulatory pathways and a dataset.

To further evaluate whether SigXTalk's performance is sensitive to experimental settings or hyperparameters, we perform additional tests using SERGIO's simulated datasets. The effects of initial learning rate and size of each training batch are investigated for hyperparameter tuning, and the effects of the number of cells, the number of genes and the levels of crosstalk are also investigated for robustness test (Supplementary Note 1).

### Implementation of other GRN-inference methods
We perform 12 representative GRN-inference methods on the simulated datasets: GENELink, PIDC, GRNVBEM, GRNBoost2, PPCOR, SCODE, SINCERITIES, LEAP, GRISLI, SINGE, Scribe and SCGSL. As they are designed for link prediction but not hyperlink prediction, we apply them to infer the activated receptor-TF link and TF-target gene link, separately. The outputs of them are the scores of pairwise gene-gene interactions at the receptor-TF and TF-target gene levels. Then, the score of a pathway in the test set is calculated by the product of absolute receptor-TF score and the absolute TF-target gene score.

GENELink is a supervised deep-learning method for reconstructing the GRNs using scRNA-seq data. It employs a graph attentional network (GAT) framework to recover the missing links that represent pairwise gene-gene interactions. Partial information decomposition for GRN inference (PIDC) uses partial information decomposition (PID) to measure multi-variable information between genes and analyze their statistical relationships. Gene Regulatory Networks based on Variational Bayesian Expectation–Maximization (GRNVBEM) employs a first-order linear autoregressive model and a variational Bayesian network to infer GRN using scRNA-data with pseudo-time ordering. GRNBoost2 uses a stochastic gradient boosting machine regression to model the expression of target genes which are regulated by the pre-

defined TFs. Partial and Part Correlation (PPCOR) infers the presence of a gene-gene interaction and constructs the weighted GRN by calculating the partial correlation between their expression levels against other variables. Single-cell ordinary differential equation (SCODE) describes the gene regulatory relationship with linear ordinary differential equations (ODEs), and the GRN is obtained by transforming and regressing these ODEs. SINgle CEll Regularized Inference using TIme-stamped Expression profileS (SINCERITIES) recovers the GRN by applying a regularized linear regression to the temporal changes in the distributions of gene expressions and identifies gene regulation mode using partial correlation analysis. Lag-based Expression Association for Pseudotime-series (LEAP) utilizes the pre-estimated pseudotime of cells and measures the co-expression strength between genes using Pearson's correlation. GRISLI infers GRN by modelling the dynamics of gene expression with a sparse regression algorithm. Single-cell Inference of Networks using Granger Ensembles (SINGE) applies a kernel-based Granger causality regression to smooth the pseudotime-resolved gene expression data, and infer the interactions between TFs and target genes by an aggregation of predictions from an ensemble of regression analyses. Scribe estimates the strength of transferred information from a regulator to its target and detects the causal relationship between genes. SCSGL learns the underlying GRN with a kernelized signed graph learning approach, where the kernels can effectively capture the nonlinear relationship between genes.

To apply GENELink to the benchmark task, the training samples are correspondingly split into two parts, so that GENELink is able to reconstruct the receptor-TF and TF-target GRN, respectively. The hyper-parameters used in the benchmark, including the learning rate, the size of neural network layers, the batch size and the number of training epochs follow the default values in the GENELink demo. For the other 11 methods, we implement them using the Beeline benchmark platform[106] with default parameters. The pseudo-time information for the SERGIO dataset required by GRNVBEM, SCODE, SINCERITIES, LEAP, GRISLI, SINGE and Scribe is generated using the Slingshot software[107], which is recommended by the BEELINE platform.

### Method evaluation using real datasets
The performance of SigXTalk is compared with NicheNet and Cyto-Talk, which both predict the downstream response to cell-cell communication, by testing whether these methods identify the shared TFs of meaningful intracellular pathways. To achieve this, we apply these methods to the HNSCC datasets where CAFs are set to the receiver cells and predict the response of differentially expressed genes in CAFs. For NicheNet, we employ two strategies for this task: (1) Niche-Net-db: directly uses the pre-constructed ligand-target matrix by selecting the targets that contain DEGs, without additional computation on the expression matrix; (2) NicheNet-comp: follows the complete workflow of NicheNet demo and calculate the ligand-target relationship using prior knowledge from NicheNet database and the gene expression profile. For CytoTalk, we follow the workflow of CytoTalk demo for the de novo construction of the signaling network from each of the other cell types to CAFs using the expression matrix and CytoTalk's built-in list of possible target genes. Since NicheNet and CytoTalk output the relationship of ligands and targets, these ligands are converted to their corresponding receptors in the CAFs. Finally, SigXTalk generates a list of all the activated receptors and TFs, which are defined as the receptor-TF pairs of the pathways with $PA > 0.75$. NicheNet-db, NicheNet-comp generate a list of all the activated receptors and targets with the regulation score $RS \geq 0.05 \times \max(RS)$. The output of CytoTalk is the final signaling network from all the other cell clusters to CAFs generated by run_cytotalk function without additional filtering.

To measure these methods' ability in identifying shared components induced by crosstalk, we compare the list of shared TFs from the prediction results and the prior ground truth database. Specifically,

the receptor-TF regulatory information from the KEGG database is extracted using the graphite R package[108] and further processed using the ToPASeq R package[109]. Through this procedure, each pathway in the KEGG database is presented as a directed graph where nodes represent genes and edges refer to pairwise gene-gene regulation. Then, we find all the possible regulatory relationships from receptors to TFs and measure their relative importance (in terms of graph topology) using the Personalized PageRank (PPR) algorithm[55]. The potential receptors are collected from the LIANA[10] and are used as the seed for the PPR algorithm. The output of the PPR algorithm is the PPR score for each receptor-TF pair. All the receptor-TF pairs with positive PPR score form the ground truth.

For the $i$-th pair of receptors, the shared TFs from method prediction results and the prior ground truth database are defined as set $A_{i,m}$ and $B_i$, respectively. The subscript $m$ represents the method (SigXTalk, NicheNet-db, NicheNet-comp and CytoTalk). The performance of the methods is measured using the following metrics:

(1) The relative coverage of the predicted shared TFs out of all the ground truth shared TFs, which could be written as:

$$\text{Coverage}_{i,m} = \#\{A_{i,m} \cap B_i\}/\#\{B_i\}. \quad (15)$$

(2) The $p$-value for the two-sided Fisher's exact test, which measures the association between the two lists when the sizes of the lists are both small. Specifically, denote $n = \#\{A_{i,m} \cup B_i\}$, $a = \#\{A_{i,m} \cap B_i\}$, $b = \#\{A_{i,m} \cap B_i\} - a$, $c = \#\{B_i\} - a$, $d = n - a + b + c$. The $p$-value of the Fisher's exact is calculated by:

$$\text{pv} = \frac{\binom{a+b}{a}\binom{c+d}{c}}{\binom{n}{a+c}}. \quad (16)$$

(3) The $p$-value for the Chi-square test, which has the similar functionality with Fisher's exact test but is suitable for larger lists. The test statistic for this test could be written as:

$$\chi^2 = \sum \frac{(O-E)^2}{E}, \quad (17)$$

where $O$ and $E$ represents the observed and expected frequency. The $p$-value of this test is estimated by comparing the statistic and the critical value: the larger the difference between observations and the expectations, the larger the Chi-square statistic is obtained, and the more significant the difference becomes.

### Estimating the activation of pathways in an individual cell

Given a target gene, the activation index (ACI) of a Signal-SSC pair quantifies how strong it is activated in an individual cell. Specifically, the ACI of the $k$-th Signal-SSC pair in the $m$-th cell is defined using the law of mass function similar to that in CellChat:

$$\text{ACI}_m^k = \frac{x_{k,m} y_{k,m}}{K_h + x_{k,m} y_{k,m}}, \quad (18)$$

where $x_{k,m}$ and $y_{k,m}$ respectively denote the expression of the $k$-th receptor and SSC in the $m$-th cell. $K_h$ is the parameter of the Hill function which is set to 0.5 as in CellChat.

### Receiver cell sub-clustering based on CCC-induced downstream regulatory events

To identify the cells that are indeed influenced by CCC events in the receiver cell cluster, we design an approach based on weighted K-means clustering. Firstly, given the pair of sender-receiver cell clusters, we calculate the communication strength for each LR pair using CellChat and remove the insignificant LR pairs. For each activated receptor (regarded as a signal) $v_{si}$, we measure the total specificity of its downstream targets and select the top 10 most specific targets as the marker genes. A binary clustering based on weighted K-means algorithm is performed using the marker genes expressed in the receiver cells, where the weight of each gene is proportional to its specificity. The outputs of the clustering are the labels for all the cells, to identify whether they participate in the communication with sender cells via the given receptor.

### Reporting summary

Further information on research design is available in the Nature Portfolio Reporting Summary linked to this article.

## Data availability

The gut lineage dataset used in this study is available in the Gene Expression Omnibus (GEO, https://www.ncbi.nlm.nih.gov/geo) under accession code GSE152325. The HNSCC dataset used in this study is available in GEO under accession code GSE103322, or at Zenodo (https://zenodo.org/records/3260758)[110]. The COVID-19 dataset used in this study is available via the single-cell portal (https://singlecell.broadinstitute.org/single_cell/study/SCP1219); note that sign-in is required to access the data. The mouse lung dataset used in this study is available in GEO under accession code GSE141259. The mouse brain spatial transcriptomics dataset can be accessed at the 10x genomics (https://www.10xgenomics.com/resources/datasets/mouse-brain-serial-section-1-sagittal-anterior-1-standard-1-0-0). Source data are provided with this paper.

## Code availability

SigXTalk is implemented on R and Python. The source code and the tutorial of SigXTalk are publicly available at the following GitHub repository: https://github.com/LithiumHou/SigXTalk and the Zenodo: https://zenodo.org/records/15453916[111]. The code that is used for reproducing the results in the article is publicly available at the following GitHub repository: https://github.com/LithiumHou/SigXTalk_scripts. The heatmap visualization in this study is developed by Gu et al[112].

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

## Acknowledgements

The work was supported by National Science Foundation grants DMS1763272, MCB2028424 and CBET2134916, a grant from the Simons Foundation (594598 to Q.N.), National Institutes of Health grants R01GM152494 and R01AR079150.

## Author contributions

Q.N., W.Z. and J.H. conceived the project; J.H., W.Z. and Q.N. designed the methods; J.H. implemented the code and performed the simulations; J.H., W.Z. and Q.N. analyzed the results; J.H. drafted the manuscript; all authors revised and approved the manuscript. W.Z. and Q.N. supervised the project.

## Competing interests

The authors declare no competing interests.
