## [Transparent Peer Review file · Nature Communications]

Dissecting crosstalk induced by cell-cell communication using single-cell transcriptomic data

Corresponding Author: Professor Qing Nie

Version 0:

Reviewer comments:

Reviewer #1

(Remarks to the Author)

The authors addressed virtually all of my comments and I believe the research community can find of interest the tool described by this study.

However, I would suggest to add and highlight in the Discussion/Limitation of the Study a paragraph describing and clearly stating how the other methods have been adapted to make them compliant with benchmark statistics that are devised for SigXTalk. The reason is that SigXTalk introduces some new metrics that the other algorithms used in the benchmarking are not optimized for. I believe that this is the reason why in the benchmarking showed with this study >90% of the methods have extremely poor performances when compared to SigXTalk. Including a paragraph discussing the limitations of these benchmarks can help readers draw more informed conclusions from the results.

Also, I would suggest to organize the Results section showing first the benchmarks across X methods on Y simulated datasets, and the same X methods on Z real-world datasets. I see discrepancies between the number/type of methods benchmarked using simulated datasets and real-world data. I also get a bit confused by the Results section showing first a case study where the method is applied (i.e., the new Figure 2), after some benchmarking, and after again other case studies. I think the manuscript flow can be improved by reorganizing the Results section by fully separating the benchmarking results (both on synthetic and real data) from the application of the method to some selected case studies where the authors highlight the relevance of the methods to novel or confirmatory biological findings.

Additional minor comments:

- 1) I think the new Figure 2 shows interesting applications of the tool. I think it would be even more interesting to see whether the authors could use SigXTalk to perform prediction on what is the actual activated pathway in each cell using only the gene expression signature of each individual cell. I believe that this would increase the utility of the method. As it is used in the new Figure 2, it seems more a descriptive use of the tool. A predictive application would be more interesting in my opinion and would let users appreciate more the unique features the tool offers.
- 2) Figure 2A and 2C share same colors although the meaning associated to the colors are different. I suggest changing the colors across the two figures so that the reader doesn't get confused (e.g., Panel 2A colors for Targets are very similar to Frizzled colors in Panel 2C and SSC colors in panel 2A are very similar to Non-canonical colors in panel 2C).
- 3) I suggest to rename Shared Signalling Component (SSC) to Transcription Factors (TFs) (or other acronyms where the role of TFs is easier to grasp, e.g., Shared Transcription Factors, sTF), if SSC entities used in all your case studies are just TFs and not other entities. If the authors have examples and case studies on real data where SSC are not TFs, then the name would make sense. But, as of now, it creates additional jargon which stays unjustified because there are no clear examples of SSC that are not TF. It will improve the clarity of the manuscript and solve any ambiguity that SSC acronym may bring (e.g., Can SSC be also signalling molecules? No, currently in this study only TFs have been modeled as SSC).

(Remarks on code availability)

The code has been reviewed by my research assistant who identified the following issues:

Issue 1: Incomplete Installation Guide

There was an installation issue when trying to set up the R dependencies required for SigXTalk. The installation guide for SigXTalk tells the users to install CellChat. This introduces a chain of missing packages that requires setting up BioManager and manual installation from Bioconductor (BiocNeighbors, Biobase, BiocGenerics, etc.). Furthermore, the scattered installation steps across multiple documentation sources provides a fragmented setup process and is not easy to follow. As a result, `devtools::install_github("jinworks/CellChat")` fails repeatedly unless the user first identifies and installs these missing packages manually — steps that is not documented in the current setup instructions.

Potential resolutions to this issue: (1) include all of its dependencies in the Remotes field of the DESCRIPTION file for automatic installation (2) adding the manual installation code in the installation guide.

Issue 2: Vignette dataset not available

The sample dataset referenced in `vignette/demo.md` is not available, and the link provided for the COVID dataset does not contain the file (broken link).

Issue 3: Incomplete data preprocessing

The code for the data preprocessing steps described in the paper, including those for the four real-world datasets outlined in Supplementary Text 2 and the filtering of clusters with fewer than 20 cells mentioned in the Methods section ("Data input and preprocessing") are not provided. The tutorial includes example code snippets, but no complete function or script for preprocessing is available, and the main code assumes that these steps have been performed externally. The pipeline also requires all cells to have predefined cluster or cell type annotations, which may limit its applicability to datasets without prior labeling.

Reviewer #2

(Remarks to the Author)

The authors have addressed my comments and the manuscript has been much improved. In my opinion, additional validation with a scMultiome ATAC+RNA data would add to the paper, however I can understand that it might be out of scope at this point.

(Remarks on code availability)

Reviewer #3

(Remarks to the Author)

Thanks to the author for responding to the comments, incorporating feedback, and significantly improving the manuscript. With the added results in real and expanded simulation datasets along with the better description, I believe the major concerns raised in the review comments are successfully addressed.

Minor comments

- Some of the added sentences can be further improved to be more clearly delivered;

ex) "For example, GENELink infers the GRN using an ..." can be rewritten to be something like "Whereas GENELink deploys the GNN based training strategy as in SigXTalk, SigXTalk outperforms them via integrating multiple upstream regulatory information"

I'm sure this will be addressed in the editing process, but please make sure the sentences read well and there is no error in the final version including:

PPR algorithm written without full abbreviation;

"Previously, ..., various mechanisms have been explored to analyze signal specificity and fidelity have been explored"

"Identifying activated regulatory pathways..." section in Method section, input of the `y_tk` MLP should be `z_tk` not `z_si`.

Gene names should be italicized.

"Together SigXTalk identifies biological -> biologically meaningful pathways.."

(Remarks on code availability)

The installation procedure is quite convoluted and I would strongly advise creating a single conda package that can be directly used without separately installing dependency, R package and Python package. This will increase the adoption of your work across labs and would increase the impact and citation of your work.

Response to Reviewers' comments

Reviewer #1 (Remarks to the Author)

1. *The authors addressed virtually all of my comments and I believe the research community can find of interest the tool described by this study.*

Response: We thank the reviewer for recognizing our efforts and the value of our study.

2. *However, I would suggest to add and highlight in the Discussion/Limitation of the Study a paragraph describing and clearly stating how the other methods have been adapted to make them compliant with benchmark statistics that are devised for SigXTalk. The reason is that SigXTalk introduces some new metrics that the other algorithms used in the benchmarking are not optimized for. I believe that this is the reason why in the benchmarking showed with this study >90% of the methods have extremely poor performances when compared to SigXTalk. Including a paragraph discussing the limitations of these benchmarks can help readers draw more informed conclusions from the results.*

Response: Thank you for your good suggestion. In the revision, we have added a paragraph to the Discussion section to discuss why other benchmarking methods show extremely poor performance, as follows:

“...Compared with SigXTalk, the other GRN inference methods do not perform well when the actual interaction from the regulator to the target has intermediate regulators by other molecules. Because the evaluation metric is designed for capturing the shared common regulators, the other general GRN inference methods naturally underperform using the such a metric. This caveat should be considered when interpreting the comparative results. Meanwhile, methods that directly measure the regulatory effect from the regulator to the target tend to overlook the specific pathways along which the regulatory signal transduces...”

3. *Also, I would suggest to organize the Results section showing first the benchmarks across X methods on Y simulated datasets, and the same X methods on Z real-world datasets. I see discrepancies between the number/type of methods benchmarked using simulated datasets and real-world data.*

Response: Sorry for the confusion. The reason for designing two different benchmark experiments for simulated datasets and real-world datasets is to address two different aspects of the SigXTalk in comparing other methods as well as due to the different

ground-truth information available to these two different kinds of datasets. More specifically,

- 1) The two benchmarking experiments allow testing two distinct functionalities of SigXTalk. The simulation data benchmark is focused on the accuracy and efficiency of the hypergraph neural networks employed in SigXTalk to identify high-order edges. For this task, GRN inference methods like GeneLink and GRNBoost2 are considered as representative ones, as they rely solely on the statistical features of the datasets rather than specific biological knowledge. On the real-world data, however, the benchmark is focused on identifying the biologically meaningful shared components induced by CCC networks. In this case, methods like NicheNet and CytoTalk provide better estimates for the effect of CCC on the expression of downstream genes, making them more natural for the validation.
- 2) The methods used in the real-world dataset (NicheNet and CytoTalk) rely on the prior (existing) knowledge of the inter- and intra-cellular interactions, which are not available to the simulation data. On the contrary, GRN inference methods used for the simulated datasets computationally produce networks of all possible gene-gene interaction pairs, without considering the biological details regarding the communication between cells and the specific pathways from receptors to target genes. Thus, such benchmarking approach is difficult to be used for estimating cells' responses to CCC and the "shared components".

To better describe the two benchmarking approaches, in the revision we have added the rationales for the choice of each benchmarking approach.

4. I also get a bit confused by the Results section showing first a case study where the method is applied (i.e., the new Figure 2), after some benchmarking, and after again other case studies. I think the manuscript flow can be improved by reorganizing the Results section by fully separating the benchmarking results (both on synthetic and real data) from the application of the method to some selected case studies where the authors highlight the relevance of the methods to novel or confirmatory biological findings.

Response: Sorry for the confusion and we appreciate your good suggestion. We admit that putting the first case study after all the benchmark experiments will make the manuscript more organized and readable. In the revision, we have re-organized the Results section based on your suggestion. The orders of the figures and the corresponding text have been changed accordingly.

Additional minor comments:

5. I think the new Figure 2 shows interesting applications of the tool. I think it would be even more interesting to see whether the authors could use SigXTalk to perform prediction on what is the actual activated pathway in each cell using only the gene expression signature of each individual cell. I believe that this would increase the utility of the method. As it is used in the new Figure 2, it seems more a descriptive use of the tool. A predictive application would be more interesting in my opinion and would let users appreciate more the unique features the tool offers.

Response: Thank you for the good point. Because a sufficient number of cells is required for the identification of activated pathways and quantification of crosstalk in SigXTalk, the inferred signaling crosstalk is at the resolution of cell clusters. In the revision, we have utilized the receptor-TF-target paths identified for cell groups and evaluate the activity of the three types of Wnt pathways (induced by canonical, frizzled and PCP receptors, respectively) for individual cells. Specifically, we evaluated the activation index of each pathway for each individual cell, and found heterogeneity in activated pathways: the canonical Wnt pathways are highly activated in ISCs, while the non-canonical pathways are highly activated in Paneth progenitors. Overall, investigating pathway activity at single-cell resolution yields results that align with the regulatory preference inferred by SigXTalk.

To illustrate the results, we have added a new supplementary figure (the new **Supplementary Figure 3**), with modified texts in the “Cell type specific crosstalk at the single-cell level” subsection in the revised manuscript:

“...The fidelity of each Wnt signal over all the possible SSCs is calculated, showing low fidelity of non-canonical in ISCs, and canonical Wnt signals in Paneth progenitors (Fig. 3c). These results are further validated, by directly estimating the activation of each pathway within each individual cell using the activation index (ACI, see Methods). Compared with non-canonical Wnt pathways, the ACI of canonical Wnt pathways are higher in ISCs, but lower in Paneth progenitors (Supplementary Fig. 3). In all, our findings agree with both results from individual-cell-based methods and previous experimental observations[53], demonstrating that the switch of Wnt pathways can be captured by SigXTalk.”

6. Figure 2A and 2C share same colors although the meaning associated to the colors are different. I suggest changing the colors across the two figures so that the reader doesn't get confused (e.g., Panel 2A colors for Targets are very similar to Frizzled colors in Panel 2C and SSC colors in panel 2A are very similar to Non-canonical colors in panel 2C).

Response: Thank you for pointing this out. We agree that the reuse of similar colors across **Figures 2A** and **2C** could lead to confusion, given the differing meanings. To

improve clarity, we have revised the color schemes in these panels to ensure that each color uniquely represents its respective category without overlap. We believe this change enhances the interpretability of the figures. Please refer to the new **Figure 3** for our revision.

7. I suggest to rename Shared Signalling Component (SSC) to Transcription Factors (TFs) (or other acronyms where the role of TFs is easier to grasp, e.g., Shared Transcription Factors, sTF), if SSC entities used in all your case studies are just TFs and not other entities. If the authors have examples and case studies on real data where SSC are not TFs, then the name would make sense. But, as of now, it creates additional jargon which stays unjustified because there are no clear examples of SSC that are not TF. It will improve the clarity of the manuscript and solve any ambiguity that SSC acronym may bring (e.g., Can SSC be also signalling molecules? No, currently in this study only TFs have been modeled as SSC).

Response: We appreciate your comments on the terminology. We admit that “SSC” does refer to “TF” in our current examples. However, we use “SSC” to emphasize the intermediate and shared molecules that may not need to be TFs, play important parts in crosstalk between pathways. The word “shared” highlights that this molecule must be affected by multiple signals or regulate multiple targets. In this context, we prefer using “SSC” for a more inclusive approach in describing the crosstalk functionality.

(Remarks on code availability)

7. The code has been reviewed by my research assistant who identified the following issues:

- Issue 1: Incomplete Installation Guide: There was an installation issue when trying to set up the R dependencies required for SigXTalk. The installation guide for SigXTalk tells the users to install CellChat. This introduces a chain of missing packages that requires setting up BioManager and manual installation from Bioconductor (BiocNeighbors, Biobase, BiocGenerics, etc.). Furthermore, the scattered installation steps across multiple documentation sources provides a fragmented setup process and is not easy to follow. As a result, ``devtools::install_github("jinworks/CellChat")`` fails repeatedly unless the user first identifies and installs these missing packages manually — steps that is not documented in the current setup instructions. Potential resolutions to this issue: (1) include all of its dependencies in the Remotes field of the DESCRIPTION file for automatic installation (2) adding the manual installation code in the installation guide.*

- *Issue 2: Vignette dataset not available: The sample dataset referenced in `vignette/demo.md` is not available, and the link provided for the COVID dataset does not contain the file (broken link).*
- *Issue 3: Incomplete data preprocessing: The code for the data preprocessing steps described in the paper, including those for the four real-world datasets outlined in Supplementary Text 2 and the filtering of clusters with fewer than 20 cells mentioned in the Methods section ("Data input and preprocessing") are not provided. The tutorial includes example code snippets, but no complete function or script for preprocessing is available, and the main code assumes that these steps have been performed externally. The pipeline also requires all cells to have predefined cluster or cell type annotations, which may limit its applicability to datasets without prior labeling.*

Response: Thank you for pointing out the issues. In the revision, we have added sufficient instructions for installing package dependencies, downloading datasets and preprocessing the data.

Reviewer #2 (Remarks to the Author)

The authors have addressed my comments and the manuscript has been much improved. In my opinion, additional validation with a scMultiome ATAC+RNA data would add to the paper, however I can understand that it might be out of scope at this point.

Response: We thank the reviewer for their positive feedback and are glad to hear that the revised manuscript has addressed the concerns raised. We agree that additional validation using scMultiome ATAC+RNA data would further strengthen the study. However, as noted by the reviewer, incorporating such data is currently beyond the scope of this work. In the revision, we have emphasized this point in the Discussion by acknowledging the value of scMultiome validation and clarifying that its integration will be an important direction for future work.

Reviewer #3 (Remarks to the Author)

1. *Thanks to the author for responding to the comments, incorporating feedback, and significantly improving the manuscript. With the added results in real and expanded simulation datasets along with the better description, I believe the major concerns raised in the review comments are successfully addressed.*

Response: We thank the reviewer for the thoughtful feedback and positive assessment. We are glad to hear that the revisions and additional results effectively addressed the major concerns.

Minor comments

2. *Some of the added sentences can be further improved to be more clearly delivered; ex) "For example, GENELink infers the GRN using an ..." can be rewritten to be something like "Whereas GENELink deploys the GNN based training strategy as in SigXTalk, SigXTalk outperforms them via integrating multiple upstream regulatory information"*

I'm sure this will be addressed in the editing process, but please make sure the sentences read well and there is no error in the final version including:

- *PPR algorithm written without full abbreviation;*
- *"Previously, ..., various mechanisms have been explored to analyze signal specificity and fidelity have been explored"*
- *"Identifying activated regulatory pathways..." section in Method section, input of the y_{tk} MLP should be z_{tk} not z_{si} .*
- *Gene names should be italicized.*
- *"Together SigXTalk identifies biological -> biologically meaningful pathways.."*

Response: We thank the reviewer for the helpful suggestion regarding clarity and phrasing. We have made revision accordingly to fix all the problems, and carefully reviewed the manuscript to ensure all additions are clearly written and free of grammatical or typographical errors.

(Remarks on code availability)

3. *The installation procedure is quite convoluted and I would strongly advise creating a single conda package that can be directly used without separately installing dependency, R package and Python package. This will increase the adoption of your work across labs and would increase the impact and citation of your work.*

Response: Thank you for your helpful suggestion. In the revised version, we have updated the package installation instructions, and we now provide simplified commands to install all required dependencies.